# Cytoplasmic LIF reprograms invasive mode to enhance NPC dissemination through modulating YAP1-FAK/PXN signaling

Shu-Chen Liu[1], Tien Hsu[1], Yu-Sun Chang[2], An-Ko Chung[3], Shih Sheng Jiang [4], Chun-Nan OuYang[2], Chiou-Hwa Yuh [5], Chuen Hsueh[6], Ya-Ping Liu[7] & Ngan-Ming Tsang[8]

Metastasis remains a clinically unsolved issue in nasopharyngeal carcinoma. Here, we report that higher levels of cytoplasmic leukemia inhibitory factor (LIF) and LIF receptor are correlated with poorer metastasis/recurrence-free survival. Further, single nucleotide variations and signal peptide mutation of LIF are identified in NPC. Cytoplasmic LIF reprograms the invasive mode from collective to mesenchymal migration via acquisition of EMT and invadopodia-associated characteristics. Higher cytoplasmic LIF enhances cancer vascular dissemination and local invasion mechanistically through modulation of YAP1-FAK/PXN signaling. Immunohistochemical analyses of NPC biopsies reveal a positive correlation of cytoplasmic LIF expression with focal adhesion kinases. Pharmaceutical intervention with AZD0530 markedly reverses LIF-mediated cancer dissemination and local invasion through promotion of cytoplasmic accumulation of YAP1 and suppression of focal adhesion kinases. Given the significant role of LIF/YAP1-focal adhesion signaling in cancer dissemination, targeting of this pathway presents a promising opportunity to block metastasis.

[1] Department of Biomedical Sciences and Engineering, National Central University, 300, Zhongda Rd., Jhongli Dist., 32001 Taoyuan City, Taiwan. [2] Molecular Medicine Research Center, Chang Gung University, 259, Wenhua 1st Rd., Guishan Dist., 33302 Taoyuan City, Taiwan. [3] Graduate Institute of Biomedical Sciences, Chang Gung University, 259, Wenhua 1st Rd., Guishan Dist., 33302 Taoyuan City, Taiwan. [4] National Institute of Cancer Research, National Health Research Institutes, 35 Keyan Rd., Zhunan, 35053 Miaoli County, Taiwan. [5] Institute of Molecular and Genomic Medicine, National Health Research Institutes, 35 Keyan Rd., Zhunan, 35053 Miaoli County, Taiwan. [6] Department of Pathology, Chang Gung Memorial Hospital at Lin-Kou, 5 Fuxing St., Guishan Dist., 333 Taoyuan City, Taiwan. [7] Pathology Core of the Molecular Medicine Research Center, Chang Gung University, 5, Fuxing St., Guishan Dist., 333 Taoyuan City, Taiwan. [8] Department of Radiation Oncology, Chang Gung Memorial Hospital and University at Lin-Kou, 5, Fuxing St., Guishan Dist., 333 Taoyuan City, Taiwan. Correspondence and requests for materials should be addressed to S.-C.L. (email: jennyliu66@gmail.com) or to N.-M.T. (email: tsangrt123@gmail.com)

Leukemia inhibitory factor (LIF) is a key component in the growth of mouse embryonic stem cells and critical regulator of embryonic development in humans[1]. Overexpression of LIF is also associated with poor prognosis in various human cancer types[2–7]. In nasopharyngeal carcinoma (NPC), LIF enhances tumor growth and is correlated with higher incidence of tumor relapse[3]. LIF activates pro-survival pathways (e.g., JAK/STAT3, PI3K, mTOR/p70S6K1, and ERK1/2) to confer cell type- or developmental stage-specific regulation of multiple biological processes, including cell proliferation, survival, and differentiation[3,8,9]. Several recent studies have shown a correlation between LIF and human cancer metastasis[7,10–13] but the mechanisms remain largely unclear. Metastasis is a multi-step process involving local extracellular matrix invasion, vascular intravasation, survival in the circulatory system, vascular extravasation, and colonization of distal organs[14]. Invadopodia are considered key structures that assist cancer cells in crossing these anatomical barriers[15]. Invadopodia modulate actin polymerization and focal adhesions (regulated by cortactin, TKS4/5, Arp2/3, cofilin, integrins), recruiting various matrix proteases (MT-MMP1, MMP2, ADAM10) to cell-matrix contacts for matrix degradation[16,17]. A number of growth factors have been shown to stimulate invadopodium formation and/or activity[18]. Several invadopodia-promoting growth factors, such as EGF, TGF-β, heparin binding (HB)-EGF, VEGF, and HGF, converge on signaling involving Src kinase, PI3K and Rho family GTPases, which control formation of invadopodia[15,19]. Pharmacological blockade of these upstream regulators of invadopodia thus presents a promising strategy to prevent metastasis.

The Hippo pathway has a crucial role in organ size control and regeneration[20]. The transcriptional coactivator, Yes-associated protein (YAP), and transcriptional coactivator with PDZ-binding motif (TAZ) function as upstream regulators of mTOR in cell size and growth control programs[21]. In human cancer, YAP/TAZ exert either oncogenic or tumor suppressor activity, depending on the cancer type and disease stage[22–25]. Roles of nuclear YAP/TAZ in regulating cytoskeleton and mechanotransduction have additionally been documented[26–28]. In breast cancer, LIFR has been identified as a tumor suppressor and a negative regulator of YAP[29]. On the other hand, LIFR has been shown to promote tumor progression in prostate cancer[30], melanoma[31], and colorectal cancer[32]. More recently, LIFR signaling has been implicated in breast cancer cell dormancy in bone marrow[33]. In the current study, we investigated the mechanisms underlying the LIF-mediated cancer metastasis and provide evidence linking LIF with cancer dissemination by driving invadopodia formation and modulation of the YAP1-FAK/PXN pathway. Moreover, our data support the therapeutic efficacy of AZD0530 (saracatinib) in suppressing vascular dissemination and local invasion in nasopharyngeal carcinoma (NPC).

## Results

### Cytoplasmic LIF and LIFR are correlated with poorer outcomes.
Previously we showed that elevated LIF in the tumor microenvironment enhances cancer radioresistance and is associated with poorer recurrence-free survival[3]. Our current findings showed the presence of LIF in nuclei of normal basal epithelia but predominant expression in the cytoplasm of tumor cells (Fig. 1a), implying diverse functional roles in normal epithelial and cancer cells. Immunohistochemical results exhibited strong immunoreactivity of cytoplasmic LIF in primary tumors obtained from NPC patients diagnosed with local recurrence or distal metastasis, which was even stronger in metastatic tumor lesions (Fig. 1b), particularly those metastasizing to liver or lung (Fig. 1b, right). Specifically, over 70% tumors metastasizing to liver or lung

expressed very high levels of cytoplasmic LIF. Furthermore, elevated cytoplasmic LIF expression was significantly correlated with poorer metastasis-free survival and recurrence-free survival ($p = 0.037$ and $p = 0.032$, respectively; log-rank test) (Fig. 1c), compared to patients with lower LIF expression. The correlations between cytoplasmic LIF levels and clinicopathological characteristics of study participants are presented in Supplementary Table 1. LIF IHC score, N stage and smoking were identified as significant adverse factors in both univariate and multivariate analyses for metastasis-free survival (Supplementary Table 2). On the other hand, LIF score was an important factor for recurrence-free survival in addition to T stage (Supplementary Table 2). Multivariate Cox regression analyses indicated that higher cytoplasmic LIF expression is an independent prognostic factor for lower metastasis-free survival [$p = 0.047$; hazard ratio = 1.873, 95% confidence interval (CI) = 1.007–3.482] and lower local recurrence-free survival [$p = 0.043$; hazard ratio = 1.838, 95% CI = 1.019–3.315] (Supplementary Table 2). Our current findings further demonstrate that higher LIFR expression is associated with poorer prognosis ($p < 0.01$, $\chi^2$ test) (Fig. 1d). The characteristics of the LIFR study population are summarized in Supplementary Table 3. Survival and univariate Cox regression analyses showed that NPC patients with higher LIFR expression (LIFR score ≥ 160) in tumors have significantly poorer metastasis-free survival ($p = 0.005$, log-rank test) and recurrence-free survival ($p = 0.015$, log-rank test) (Fig. 1e) vs. patients with lower LIFR levels in tumors. Multivariate Cox regression analyses revealed that higher LIFR expression is an independent prognostic factor for lower metastasis-free survival [$p = 0.006$; hazard ratio = 2.704; 95% CI = 1.322–5.532] and lower local recurrence-free survival [$p = 0.023$; hazard ratio = 2.198; 95% CI = 1.117–4.327] (Supplementary Table 4). Our findings clearly indicate that higher cytoplasmic LIF and LIFR expression are adverse predictors for metastasis-free and recurrence-free survival. Correlation analyses revealed a close correlation between cytoplasmic LIF and LIFR expression ($p < 0.0001$; $r = 0.4036$; Spearman's correlation test) (Fig. 1f, g). Taken together, these results demonstrate that higher cytoplasmic LIF and LIFR levels are associated with poorer outcomes of NPC patients.

### Spectrum of single nucleotide variation of LIF in NPC.
Canonical LIF can exist as a secreted cytokine or be retained in the intracellular compartments[34–36]. Receptor downregulation/desensitization of IL-6 family cytokines is a mechanism for signal attenuation in response to repeated or chronic ligand stimulation[37]. The fact that both LIFR and LIF are overexpressed in NPC tumors derived from patients with metastasis implies that elevated cytoplasmic LIF in tumor cells may not originate from the microenvironment. One of the potential causes leading to increased intracellular LIF level in NPC is genetic alterations. Among them, signal peptide mutations have been reported to result in accumulation of cytoplasmic cytokines, leading to receptor self-association and increased constitutive signal transduction[38,39]. To determine whether the genetic alterations of LIF were present in NPC, we conducted LIF deep-sequencing (Illumina) on 50 formalin-fixed paraffin-embedded (FFPE) NPC samples as well as Sanger sequencing on 107 FFPE NPC samples (see Supplementary Methods) and identified a variety of single nucleotide polymorphism (Fig. 1h). Two missense mutations were detected in NPC, one located within region of signal peptide (G20L) and the other was found in exon 3 (P90L) (Fig. 1h). Interestingly, the IHC data showed that NPC tumors derived from patient harboring LIF signal peptide mutation expressed higher levels of LIF and LIFR (Fig. 1i). We further systematically analyzed available cancer mutation data from cBioportal database

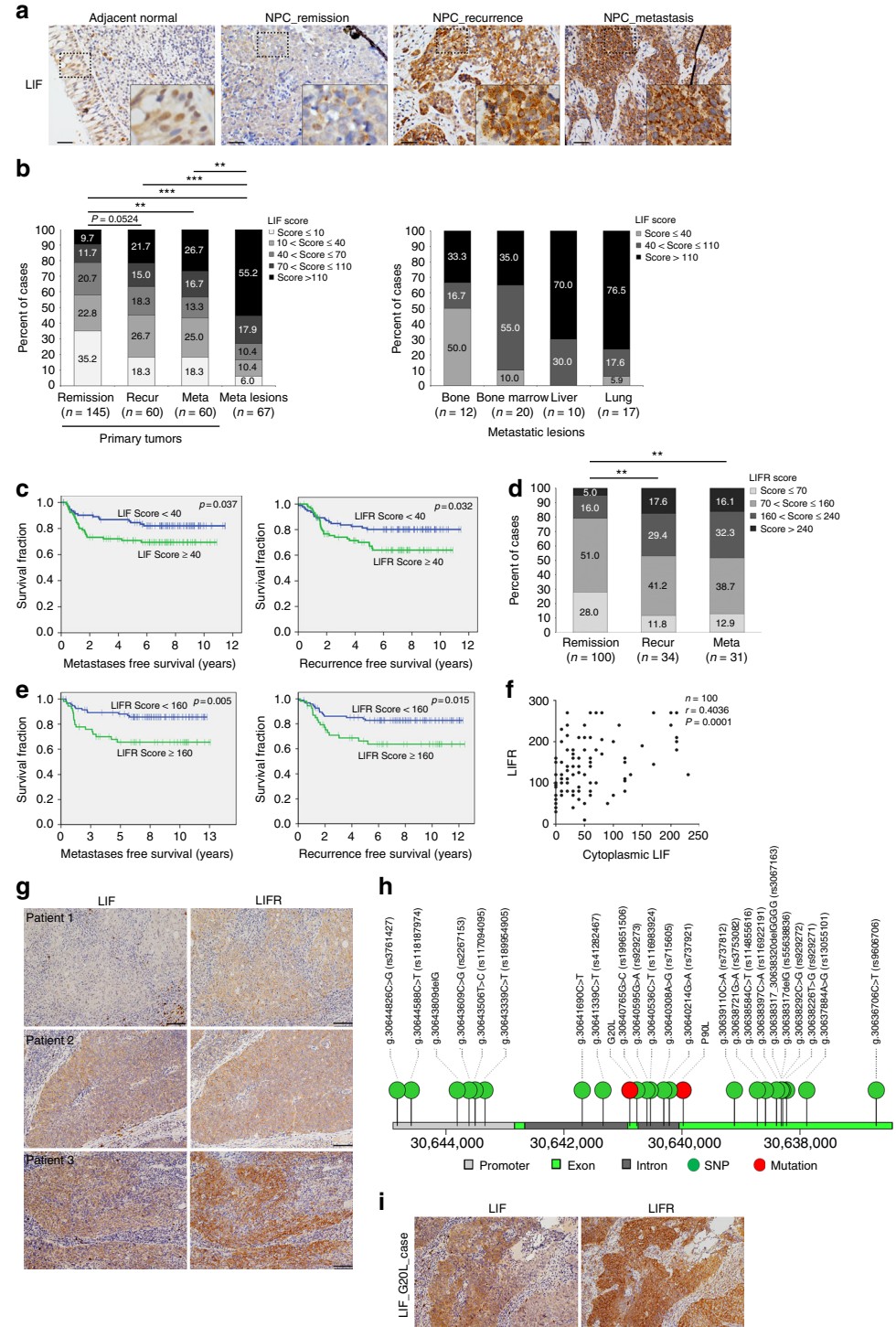

**Fig. 1** Elevated cytoplasmic LIF and LIFR in NPC are correlated with poorer prognosis. **a** Representative images of LIF expression in adjacent normal epithelium and NPC tumor tissues. Scale bars, 20 μm. **b** Statistical analysis of cytoplasmic LIF expression in primary NPC tumor tissues and metastatic lesions. Analysis of cytoplasmic LIF expression in distinct metastatic lesions is shown (right). **p < 0.01, ***p < 0.001, chi-square test. **c** Kaplan–Meier survival curves of NPC patients based on IHC scores of cytoplasmic LIF expression. Metastasis-free survival (left). Recurrence-free survival (right). **d** Statistical analysis of LIFR expression in NPC tumor biopsies. **p < 0.01, chi-square test. **e** Kaplan–Meier survival analysis of NPC patients based on LIFR expression. Metastasis-free survival (left). Recurrence-free survival (right). **f** Analysis of correlation between cytoplasmic LIF and LIFR expression (Spearman's correlation test). **g** Representative images of LIF and LIFR expression patterns in NPC tumor tissues. Scale bars, 50 μm. **h** Single nucleotide variations in NPC biopsy samples (n = 157). Circles are colored with respect to the corresponding mutation types. **i** LIF and LIFR expression in tumors from NPC patient carrying LIF signal peptide mutation (G20L). Scale bars, 50 μm

(216 studies)[40] and found that a relatively high frequency of LIF gene mutations (>2%) were detected in melanomas, lung small cell carcinomas, uterine cancers, and cholangiocarcinoma (Supplementary Figure 1a, green bars). The mutation sites are distributed across the coding regions with a higher incidence in exon 1 and exon 3 as shown in Supplementary Figure 1b. The mutation types mainly include missense, truncating, and inframe mutations (Supplementary Table 5). Noticeably, two signal peptide mutations were detected in skin cutaneous melanomas (W17L and G22W). In addition, the mutation of P90L identified in our NPC samples, was also found in colorectal adenocarcinoma and stomach adenocarcinoma (Supplementary Figure 1b and Supplementary Table 5). Whether these mutations are associated with tumor progression will need further experimental and clinical validations.

**Characterization of established LIF mutant cancer clones**. To determine the precise role of intracellular LIF in NPC invasiveness, we utilized TALEN technology to generate stable clones with mutations in the signal peptide region (cLIF clones) to induce cytoplasmic accumulation of LIF or with loss of the initiating codon in one allele (LIF$^{+/-}$ clone) to reduce LIF expression (Fig. 2a; Supplementary Figure 2). Western blot analyses showed higher LIF expression in cLIF cells and lower expression in LIF$^{+/-}$ cells, compared to LIF-WT parental cells (Fig. 2b). Results of cytokine assays (Bio-Rad) showed that WT cells secreted higher amount of LIF (mean ± SEM (pg/ml): 37.7 ± 1.3) into medium whereas the cLIF cells secreted little amount of LIF (6.9 ± 1.5) into culture supernatant. The LIF level secreted by the LIF$^{+/-}$ cells was about half of the WT (17.6 ± 1.6) (Fig. 2c). Immunocytochemistry findings confirmed the presence of strong cytoplasmic LIF signals in cLIF cells in contrast to reduced intracellular LIF in LIF$^{+/-}$ cells (Fig. 2d). Established LIF mutant clones also exhibited morphological differences (Fig. 2e). Parental cancer cells exhibited a cobblestone shape whereas cLIF cells showed a spindle-like morphology. In contrast, LIF$^{+/-}$ cells displayed a flattened shape (Fig. 2e). To evaluate cellular response to exogenous LIF stimulation, we treated cancer cells with recombinant human LIF pre-labeled with ATTO 488 and found that cLIF cells appeared resistant to the uptake of LIF, unlike that observed in WT or LIF$^{+/-}$ cells (Fig. 2f). We further evaluated LIFR expression and its response to LIF stimulation in these cell strains. Results showed that cLIF cells expressed abundant LIFR whereas LIF$^{+/-}$ cells contained low levels of LIFR, compared to their WT counterparts (Fig. 2g). Moreover, results of time-course experiments on cells treated with recombinant human LIF showed that in WT cancer cells, LIFR desensitization occurred at 30 min and is sustained up to 2 h post-LIF treatment. In cLIF cancer cells, the LIFR level was constitutively high, indicating they were not responsive to exogenous LIF stimulation. On the other hand, LIF$^{+/-}$ cells exhibited little response to LIF stimulation, probably as a result of very low basal LIFR levels (Fig. 2g). Moreover, LIF stimulation activated downstream key kinase p70S6K1 (T389) in WT cells and to a much less extent in LIF$^{+/-}$ cells (Fig. 2g). The cLIF cells expressed higher level of activated p70S6K1 (T389), which was only slightly enhanced by LIF stimulation. These findings indicate that LIF can regulate LIFR expression, consistent with the patterns presented in clinical NPC biopsy samples (Fig. 1f, g).

**Higher cLIF reprograms the cancer invasion mode**. Previous immunohistochemical analyses showed that higher cytoplasmic LIF is significantly correlated with NPC metastasis (Fig. 1a–c). Cellular LIF-high cancer cells (cLIF) exhibited fibroblastic morphology (Fig. 2d, e), leading to the suggestion that cytoplasmic

LIF reprograms the cancer invasion mode. To examine this hypothesis, we initially measured cell migration using time-lapse live imaging in wound-healing experiments. Both WT parental and LIF$^{+/-}$ cancer cells exhibited a collective migration pattern whereas cLIF cells displayed a mesenchymal migration mode (Fig. 3a; Supplementary movies 1–3). To further evaluate the invasion ability of cancer cells, we conducted 3D-gelatin matrix invasion assays. Results demonstrated that the cLIF cells could invade into deeper depth compared with WT or LIF$^{+/-}$ cells (38 vs. 10 μm) after 5 h post-plating (Fig. 3b). Moreover, we generated mouse xenograft tumors and evaluated the extent of tumor mass invasion into the surrounding muscle and stromal tissues via analysis of hematoxylin and eosin staining of tumor sections. Calculation of the events of local tumor invasion divided by the total number of mouse xenografts examined under the microscope revealed that cLIF tumors displayed a higher frequency of invasion into nearby muscle tissues, compared to that of WT parental cells at 4 weeks post-inoculation (76.9% vs. 55.6%) (Fig. 3c, d). In contrast, LIF$^{+/-}$ tumors showed a lower frequency of invasion into local tissues and formed smaller tumor buds. LIF-induced invasiveness of cLIF tumors was correlated with stimulation of the epithelial–mesenchymal transition (EMT) process, as evident from the increased expression of EMT markers N-cadherin (N-cad), vimentin (VIM), and IQ Motif Containing GTPase Activating Protein 1 (IQGAP1) and decreased expression of E-cadherin (E-cad) (Fig. 3e). LIF$^{+/-}$ cells displayed the opposite expression patterns for these markers, suggesting EMT inhibition. Invadopodia are considered key components of cancer cells that may aid in penetrating anatomical barriers[15,41]. Western blot results showed that expression levels of invadopodia markers, such as tyrosine kinase substrate with five SH3 domains (TKS5), cortactin (CTTN), matrix metallopeptidase 2 (MMP2), and the upstream regulator, SRC proto-Oncogene (SRC), were increased in cLIF and decreased in LIF$^{+/-}$ cells, compared to WT parental cells (Fig. 3e). Immunostaining experiments revealed that higher cytoplasmic LIF leads to the remodeling of actin organization resulting in an actin bundle-rich, long protrusion phenotype in cLIF cells (Fig. 3f–h). In addition, TKS5 and CTTN were more widely distributed in the cytosol and abundant at the migrating fronts of cLIF cells, whereas TKS5 and CTTN expression were observed at the belly side (near the nucleus) of WT and LIF$^{+/-}$ cells (Fig. 3f, g). Additionally, enhanced MMP2 expression was evident at the migrating fronts and extracellular matrices of cLIF cells (Fig. 3h). Taken together, the data suggest that cytoplasmic LIF is likely to be involved in matrix degradation to facilitate invasion into nearby tissues, which is potentially achieved by inducing EMT and invadopodia formation.

**Higher cytoplasmic LIF enhances cancer vascular invasion**. Recent research has reported a critical role of invadopodia in cancer vascular invasion and distal metastasis[42]. To evaluate whether higher cytoplasmic LIF facilitates vascular invasion, we initially utilized the real-time impedance cell analyzer to assess cancer-induced disruption of a confluent endothelial layer (HUVEC). Cancer cell-mediated reduction of impedance was evaluated within 6 h after addition of cancer cells to avoid the influence of cell replication. Seeding of cLIF cells on the HUVEC layer led to a more rapid drop in impedance compared to that caused by WT parental or LIF$^{+/-}$ cells (Fig. 4a). The magnitude of decreased impedance was proportional to the number of seeded cancer cells. We evaluated the disrupted areas of HUVEC via double immunofluorescent labeling for VE-cadherin (endothelial marker) and pan-cytokeratin (carcinoma marker) in one-day co-cultured cells. cLIF cells were capable of creating larger holes in the HUVEC layer than WT cancer and LIF$^{+/-}$ cancer cells

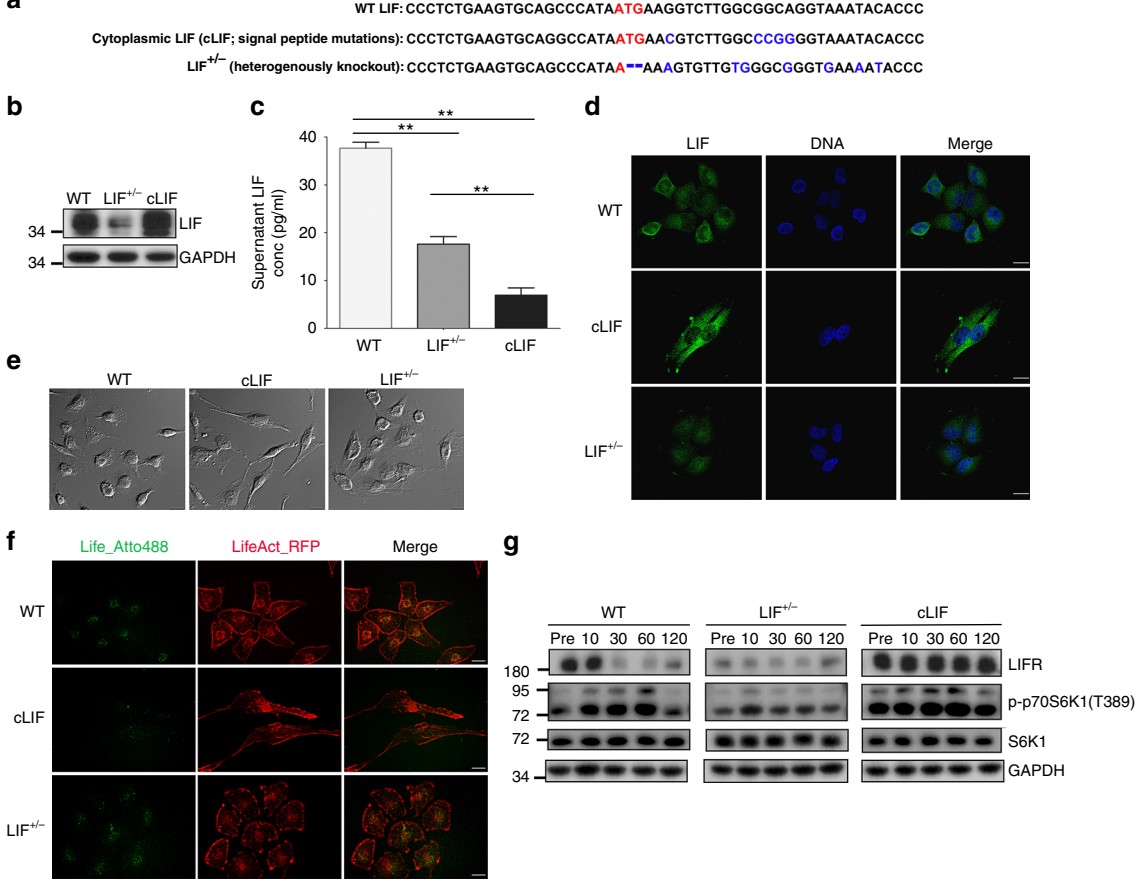

**Fig. 2** Characterization of LIF mutant clones. **a** Sequence analysis of the LIF gene. Genomic DNA was extracted from parental NPC BM1 cells with wild-type LIF or established clones either with mutations in the signal peptide region of LIF (cLIF clone) or loss of the initiating codon in one allele (LIF$^{+/-}$ clone). The initiating codon within the spacer is indicated in red. Mutated nucleotides are marked in blue. **b** Assessment of LIF protein expression via western blot using GAPDH as a loading control. **c** Assessment of secreted LIF using a bead-based cytokine assay. Supernatants were harvested 2 days post culture. Data are presented as means ± SD of triplicate experiments. **p < 0.01, two-tailed, paired t test. **d** Immunofluorescent detection of LIF (green) in WT, cLIF and LIF$^{+/-}$ cancer cells. Blue, nuclear staining. Scale bars, 10 μm. **e** Comparison of morphological changes (DIC images) in WT, cLIF, and LIF$^{+/-}$ cancer cells. Scale bars, 10 μm. **f** Live images of LIF uptake in cancer cells expressing LifeAct-RFP. Recombinant LIF proteins were pre-labeled with ATTO 488 green fluorescent dye. Images were captured 40 min post-LIF addition. **g** Time-course analysis of LIFR desensitization and p70S6K1 activation in LIF (30 ng/ml)-stimulated cells using GAPDH as a loading control

(Fig. 4b). Quantitative analyses further disclosed that cLIF cells generate a two-fold increase in the invaded areas, compared to damage by WT or LIF$^{+/-}$ cells (Fig. 4c). To further characterize the mechanism by which intracellular LIF modulates breakdown of endothelial barriers, we utilized high-resolution time-lapse fluorescence microscopy, which provided spatial and temporal information on the interactions between cancer and HUVEC cells. To this end, we seeded equal numbers of cancer cells expressing LiveAct-RFP into cultures of confluent HUVEC cells expressing LiveAct-GFP2 for live-cell imaging. As shown in Fig. 4d, LIF-rich cancer cells (cLIF) caused more significant damage to the HUVEC layer, which appeared unrepairable, compared to that induced by WT parental and LIF$^{+/-}$ cancer cells (Supplementary movies 4-6). Moreover, cLIF cells showed evident downregulation of endothelial junctional markers (CD31 and VE-cadherin) along the borders of the disrupted area, assessed via immunofluorescence staining (Fig. 4e). A single cLIF cell was capable of generating elongated protrusions and disrupting a wide range of endothelial junctions. Data from western blot analyses also showed decreased expression of CD31 and VE-cadherin in cLIF cells (Fig. 4f), but not WT and LIF$^{+/-}$ groups. This finding may be attributable to the fact that only a limited number of cancer cells were co-cultivated with HUVEC cells.

Consequently, WT and LIF$^{+/-}$ cancer-induced damage was relatively minor and could be repaired by HUVEC cells. The functional importance of our in vitro results was further validated in the zebrafish model in vivo. Transparent zebrafish embryonic xenografts provide a reliable model to analyze metastasis of human cancers[43]. The yolk sac of zebrafish embryos is an acellular and nutrient-rich environment where cancer cells can grow and migrate to distal sites through the circulation. We established zebrafish embryonic xenograft models by injecting an equal number of cancer cells expressing LifeAct-RFP into the yolk sac of 48 h post-fertilization (hpf) embryos of Tg(fli1a:EGFP)y1, an EGFP$^+$—endothelial transgenic zebrafish line, and measured vascular dissemination of tumor-like structures at 6 days post-injection. We set the cutoff value of the tumor area at ≥10 μm$^2$ (approximately ≥3 cells) to indicate growth of disseminated tumor-like structures in zebrafish embryos (Fig. 4g, h). Quantification of vascular disseminated tumor-like structures revealed the presence of a greater number of structures in embryos injected with cLIF cancer cells and fewer structures in embryos injected with LIF$^{+/-}$ cells, compared to those injected with WT cancer cells [mean (SD): WT: 37.3 (16.9), cLIF: 55.9 (17.4), LIF$^{+/-}$: 24.1 (8.0)] (Fig. 4i). These data collectively support a critical role of LIF in cancer vascular evasion.

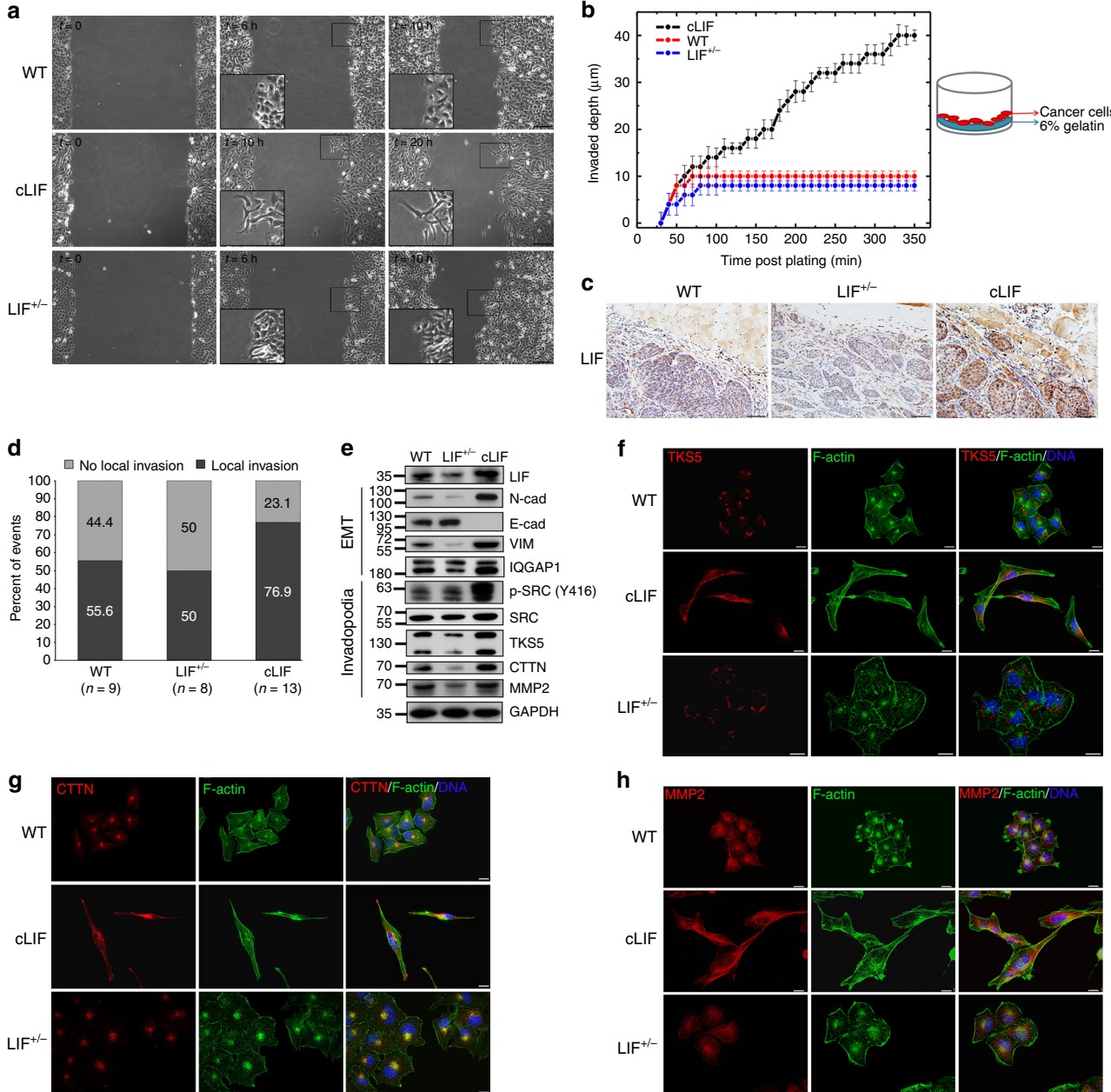

**Fig. 3** Cytoplasmic LIF regulates EMT and invadopodia formation. **a** Migration pattern analysis using time-lapse live imaging in wound-healing experiments. Scale bars, 100 µm. See also Supplementary movies 1–3. **b** 3D-gelatin invasion assays. The invasion depth of cancer cells through a stiff gelatin matrix (6%) were measured by time-lapse phase contrast vertical scanning (Olympus IX83). Values are presented as means ± SD of triplicate experiments. **c** Representative IHC images of LIF expression in paraffin-embedded mouse tumor xenografts. Scale bars, 50 µm. **d** Quantification of mouse tumor xenografts with events of local invasion based on hematoxylin & eosin (HE) staining analysis. **e** Western blot of EMT and invadopodia markers in wild-type parental, cLIF and LIF$^{+/-}$ cancer cells. GAPDH was used as a loading control. **f–h** Immunofluorescence staining of invadopodia markers: TKS5 (**f**). CTTN (**g**). MMP2 (**h**). Alexa Fluor 488 phalloidin (green) was used to stain F-actin. Blue, nuclear staining. Scale bars, 10 µm

**cLIF activates focal adhesion kinases.** Invadopodia and focal adhesions are extensively characterized as hallmarks of cancer metastasis[41,44]. Here, we aimed to investigate whether LIF modulates cancer adhesion to facilitate metastatic dissemination. To this end, we first evaluated the time of cancer cell attachment on the surface of culture dishes. Results of live imaging analyses showed that cLIF cells exhibit spreading appearance at ~6 h postplating, similar to images acquired at 24 h (Supplementary Figure 3). On the other hand, many of the WT and LIF$^{+/-}$ cancer cells still exhibited a round-up morphology at 6 h, although they displayed spreading at 24 h post-plating indicative of a less

efficient attachment mechanism in these cell groups. Since cell attachment is tightly regulated by focal adhesion molecules, we examined the expression patterns of two key focal adhesion kinases, Paxillin (PXN) and Focal adhesion kinase (FAK), in these cell strains. cLIF cells expressed higher levels of activated PXN [p-PXN (Y118)] and FAK [p-FAK (Y397)], whereas LIF$^{+/-}$ cells exhibited lower levels of these molecules, compared to WT cells (Fig. 5a). Abundant p-PXN and p-FAK were detected at the migrating fronts of cLIF cells while primarily presenting at the peripheries of LIF-WT and LIF$^{+/-}$ cells (Fig. 5b). Furthermore, expression of focal adhesion kinases was associated with

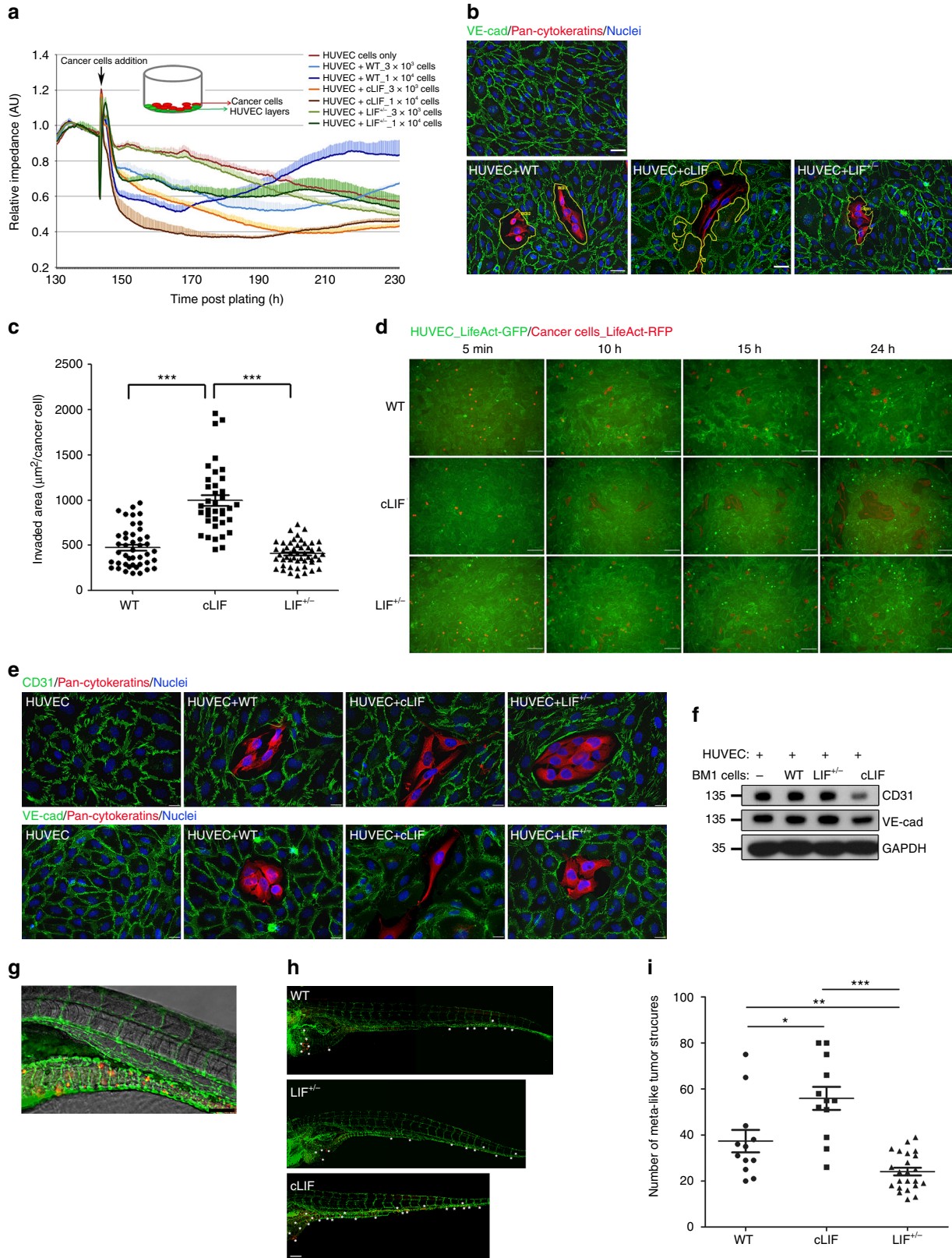

formation of actin bundles in cLIF cells. To better understand the mechanisms linking focal adhesion molecules to transendothelial invasion of cancer, we pre-labeled cancer cells with Talin-GFP before plating on top of the HUVEC layer and monitored expression of focal adhesion molecules in live cells. Talin signals were extensively distributed at the protrusions of cLIF cells

whereas hot spots were primarily detected at the belly of WT and LIF$^{+/-}$ cells (Fig. 5c). Notably, damaged HUVEC areas were detected at the invasive front with high Talin-GFP expression (Fig. 5c, white arrow). To evaluate whether cytoplasmic LIF is correlated with focal adhesion molecules in human tumors, we examined the expression patterns of LIF, p-PXN (Y118), and p-

**Fig. 4** High cytoplasmic LIF enhances cancer vascular invasion. **a** Real-time impedance analysis. HUVEC cells were grown on E-plates until confluence. Cancer cells were added on top of the HUVEC layer at the indicated times (black arrow). data are presented as means and SD of triplicate experiments. **b** Representative images of HUVEC layer replacement assay. Equal numbers of cancer cells were plated on top of the confluent HUVEC layer and co-cultivated for 24 h. Cells were labeled with antibodies against pan-cytokeratin (red) and VE-cadherin (green). Yellow closed polygons indicate damaged HUVEC areas. Scale bars, 20 μm. **c** Quantification of displaced areas in **b**. Data are presented with scatter dot plot (mean ± SEM). Each black dot represents one captured image. Invaded areas were calculated using CellSens imaging software (Olympus). ***$p < 0.001$, Mann–Whitney test. **d** Live-cell imaging of the HUVEC replacement assay. Cancer cells stably expressing LiveAct-RFP were plated onto confluent HUVEC cells expressing LiveAct-GFP2. Live interactions were continuously monitored for 30 hours (see also Supplementary movies 4-6). **e** Addition of cancer cells to the HUVEC layer breaks down endothelial junctions. An equal number of cancer cells was added on the top of HUVEC layer and co-cultivated for 24 h before fixation. Cells were labeled with antibodies against pan-cytokeratin (red) and CD31 (green, upper panels) or pan-cytokeratin and VE-cadherin (green, bottom panels). Scale bars, 10 μm. **f** Western blot of CD31 and VE-cadherin expression using GAPDH as a loading control. **g** Representative image of Tg (fli1a:EGFP)y1 zebrafish embryo carrying cancer cells expressing LifeAct-RFP on day 6 post-cancer cell injection. Tumor-like structures (red) that spread through the vasculature (green) were observed. **h** Representative images of disseminated tumor-like structures in zebrafish embryos injected with WT, LIF$^{+/-}$, or cLIF cells. Asterisks indicate the dissemination sites of tumor-like structures. **i** Quantification of tumor-like structures shown in **h**. Areas of tumor dots were calculated using CellSens imaging software. The number of counts was determined 6 days after injection with cancer cells. Data are presented with scatter dot plot (mean ± SEM). Each black dot represents one fish embryo. *$p < 0.05$, **$p < 0.01$, ***$p < 0.001$, Mann–Whitney test

FAK (Y397) in consecutive NPC biopsy sections. IHC results demonstrated a positive correlation of cytoplasmic LIF expression with p-FAK ($p = 0.0135$, $r = 0.3618$) and p-PXN ($p = 0.0612$, $r = 0.2694$, Spearman's correlation test) (Fig. 5d–f). Our findings collectively indicate that cytoplasmic LIF positively regulates focal adhesion dynamics to facilitate cancer invasion.

**LIF promotes NPC invasiveness via the LIFR/YAP pathway**. The regulation of YAP1 localization is critical for maintaining density-dependent cell-cell junctions and mechanotransduction[27]. The binding of 14-3-3 to the phosphorylated YAP1 (S127) has been linked to cytoplasmic retention and consequent loss of the transcription factor function of YAP1 protein[45]. In our cell model, LIF/LIFR-rich cancer cells (cLIF) expressed lower levels of phosphorylated YAP1 as well as total YAP1 protein (Fig. 6a, b), leading to the hypothesis that LIF regulates LIFR–YAP1 signaling to promote NPC invasive phenotypes. To examine this theory, LIFR was depleted in WT and cLIF cancer cells expressing medium to high levels of cellular LIF and LIFR, respectively. Knockdown of LIFR led to increased phosphorylation of YAP1 at S127 in cLIF cells (Fig. 6c). Importantly, depletion of LIFR also decreased level of activated SRC, implying the role of LIFR in LIF-mediated NPC metastasis. The relationship between YAP1 function and NPC metastasis was further examined. WT and LIF$^{+/-}$ cancer cells expressed medium to high levels of YAP1. Surprisingly, loss of YAP1 expression led to dramatic alterations in phenotype and cancer behavior. Depletion of YAP1 increased activations of focal adhesion molecules (PXN (Y118), FAK (Y397), and SRC (Y416)) (Fig. 6d). Results of immunostaining revealed morphological changes from a flattened epithelioid appearance to spindle-like fibroblastic morphology in YAP1-depleted WT and LIF$^{+/-}$ cells (Fig. 6e; Supplementary Figure 4), resembling that observed in cLIF cancer cells. We further conducted the HUVEC layer replacement assay to evaluate the invasive ability of YAP1-depleted cells. Immunostaining data clearly showed that YAP1-depleted cancer cells exhibit longer protrusions on HUVEC layers (Fig. 6f). Quantification of the disrupted areas of HUVEC layers revealed that loss of YAP1 leads to greater HUVEC damage (Fig. 6g, h). Immunohistochemical analyses revealed an inverse correlation between expression of LIFR and cytoplasmic YAP1 in some primary NPC tumors and those metastasizing to bone marrow (Fig. 6i). This finding is consistent with lower expression of p-YAP1 and YAP1 in cLIF cells (high invasiveness), compared to higher expression in WT LIF and LIF$^{+/-}$ cells (lower invasiveness) (Fig. 6a). These results support the theory that LIF-LIFR signaling is critical for

metastatic behavior in NPC cells and linked to suppression of phosphorylated YAP1 (S127).

**AZD0530 suppresses LIF-mediated cancer invasion**. The small molecule, AZD0530, is a dual inhibitor of the Src tyrosine kinase family and ABL. Since cLIF cells exhibited a higher activity of SRC and consequently an enhanced EMT and invadopodia formation (Fig. 3), we therefore investigated whether treatment of cancer cells with AZD0530 alters LIF-regulated expressions of p-YAP1 and focal adhesion molecules, which might affect SRC activity. To evaluate the cellular responses to and determine the IC$_{50}$ values of cancer cells to AZD0530, we treated cancer cells with a wide range of AZD0530 doses and monitored cellular responses using a real-time impedance analyzer. Results demonstrated that the cLIF cells were more resistant to AZD0530 treatment (IC$_{50}$ = $1.73 \times 10^{-5}$ M) (Supplementary Figure 5c, d) whereas the LIF$^{+/-}$ cells were more sensitive to AZD0530 treatment (IC$_{50}$ = $4.53 \times 10^{-7}$ M) (Supplementary Figure 5e, f), compared with LIF wild-type cells (IC$_{50}$ = $8.93 \times 10^{-7}$ M) (Supplementary Figure 5a, b). Western blot analyses revealed that treatment of cancer cells with AZD0530 led to increased expression of p-YAP1 (S127) in cLIF and WT cells. On contrary, the expression of p-YAP1 in LIF$^{+/-}$ cells was less affected by AZD0530, probably due to the low levels of LIFR and LIF presented in LIF$^{+/-}$ cells. Exposure of AZD0530 also decreased expression of activated focal adhesion molecules (p-PXN and p-FAK) (Fig. 7a). Immunocytochemical data revealed that AZD0530 promotes cytoplasmic YAP1 accumulation and loss of nuclear expression in cancer cells (Fig. 7b). We further evaluated whether AZD0530 suppresses local tumor invasion in mouse tumor xenografted model. Results of IHC analysis showed that AZD0530 treatment promoted cytoplasmic YAP1 accumulation along with decreased expression of activated p-PXN (Y118) and p-FAK (Y397) in both WT and cLIF tumors (Fig. 7c–e). IHC results further demonstrated that AZD0530 could efficiently prevent tumor local invasion. Local invasion rates were decreased from 62.5 to 20% in WT tumors and 83.3 to 16.7% in cLIF tumors (Fig. 7f). To evaluate whether AZD0530 affects cancer vascular dissemination, we conducted the HUVEC layer replacement assay. Under the co-cultured system, AZD0530 treatment led to a reduced cancer invadopodia formation, compared to the vehicle-treated group (Fig. 8a). Quantification results of HUVEC layer displacement assays revealed that treatment with AZD0530 (5 μm) reduced the damaged area to ~50%, compared to that observed in the vehicle control groups [for cLIF cells, vehicle vs. AZD0530; mean (SD): 669.4 (221.7) vs. 371.2 (113.6)] (Fig. 8b–d). Western blot data showed that treatment with

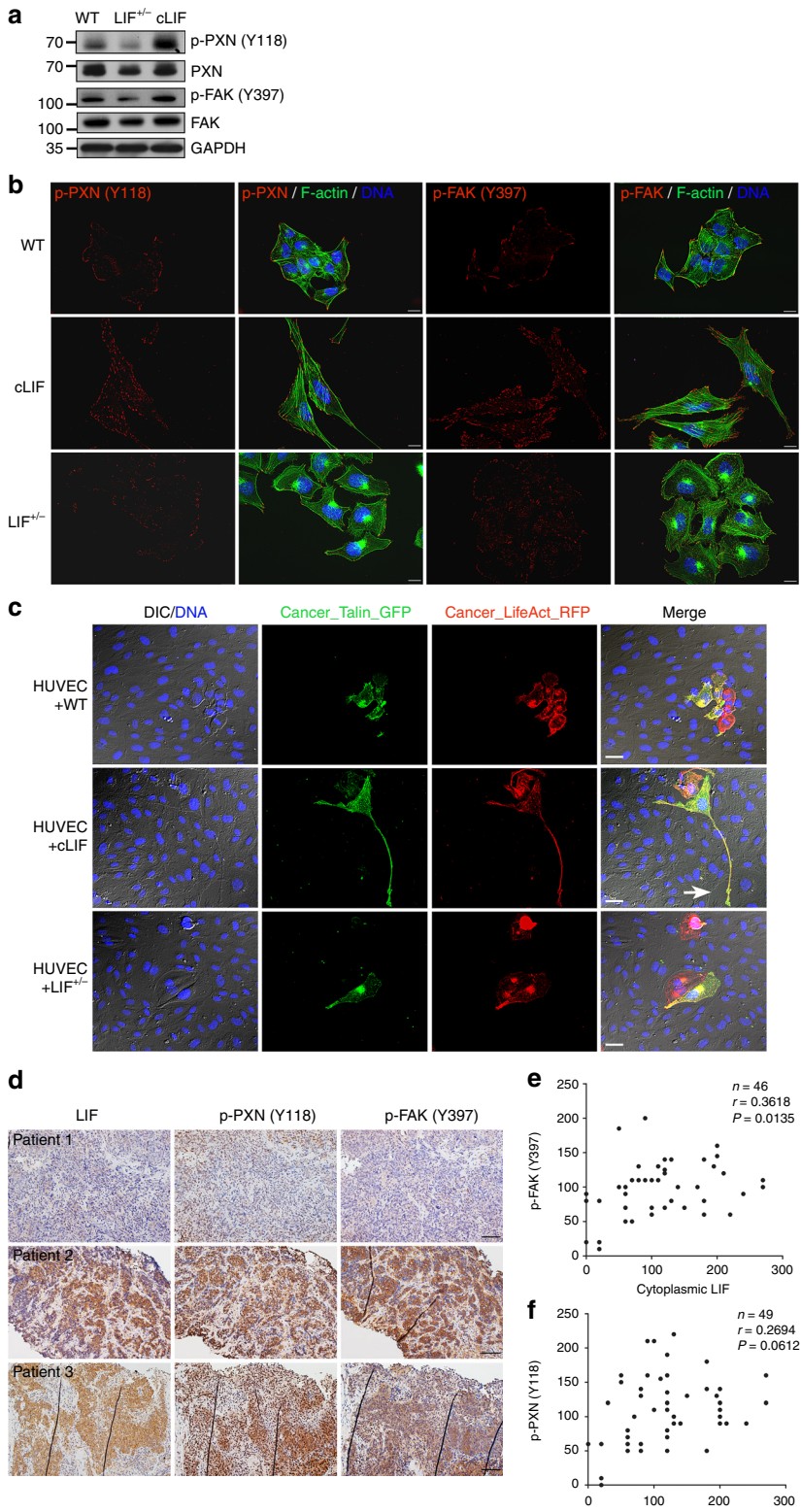

**Fig. 5** LIF regulates focal adhesion molecules. **a** Detection of endogenous focal adhesion kinases in cancer cells via western blot using GAPDH as a loading control. **b** LIF regulates the spatial distribution of activated focal adhesion kinases. Cells were labeled with antibodies against phospho-PXN (Y118) or phospho-FAK (Y397). Alexa Fluor 488 phalloidin (green) was used to stain F-actin. Blue, nuclear staining. Scale bars, 10 μm. **c** Live imaging of focal adhesion during transendothelial invasion. Cancer cells expressing LifeAct-RFP were pre-labeled with Talin-GFP and plated onto the HUVEC layer. Images were captured 24 h post-plating. Blue, nuclei labeled with Hoechst33342. Scale bars, 20 μm. The white arrow indicates the damaged area caused by Talin-rich elongated protrusion. **d** Representative images of LIF, p-PXN (Y118), and p-FAK (Y397) expression in paraffin-embedded consecutive NPC tissue sections. Scale bars, 50 μm. **e, f** Correlation analyses based on IHC scores (Spearman's correlation test). Correlations were evident between LIF and p-FAK (Y397) (**e**) and LIF and p-PXN (Y118) (**f**)

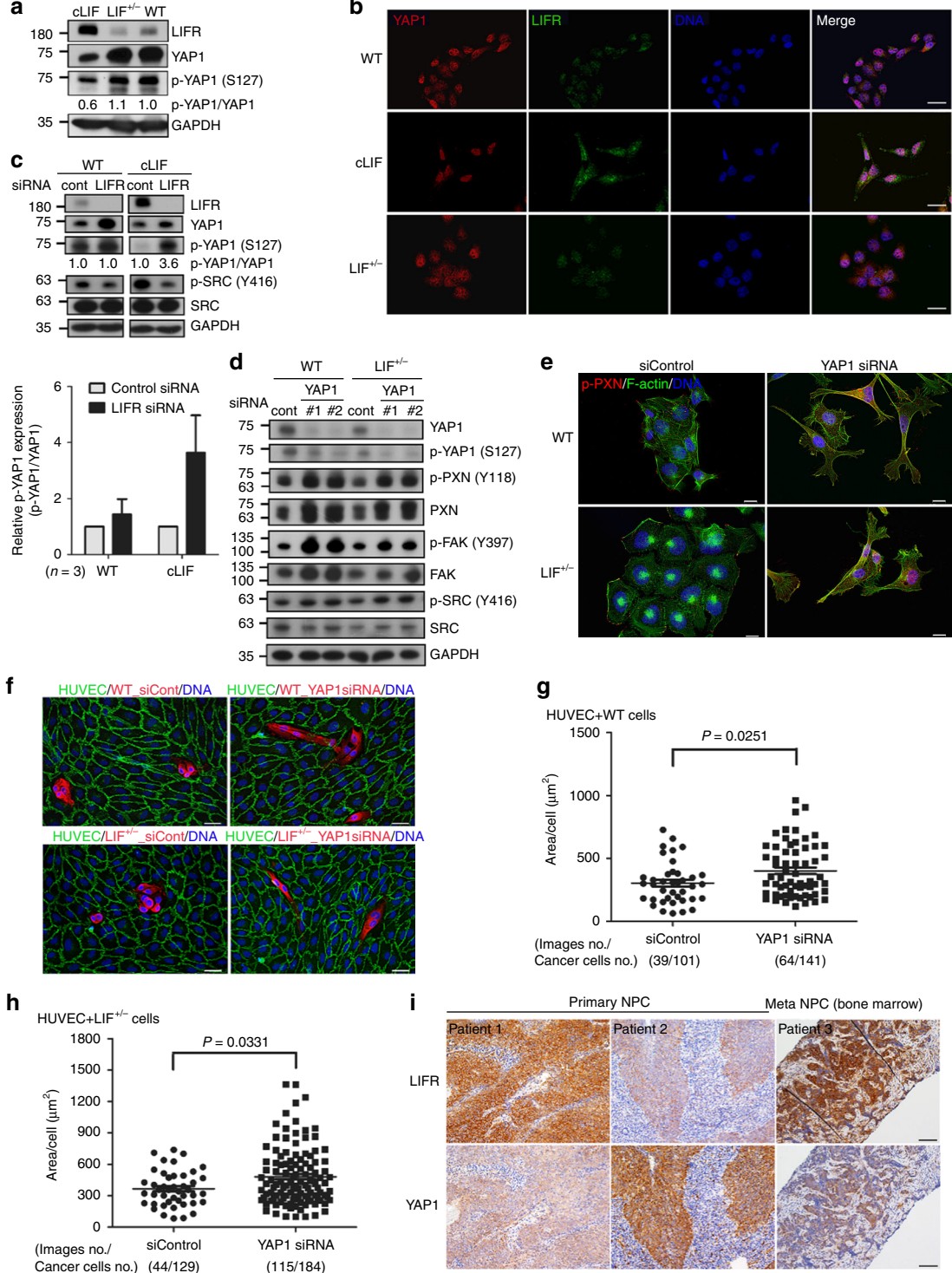

**Fig. 6** LIFR–YAP1 signaling is critical for LIF-mediated invasion of NPC cells. **a** Endogenous protein expression of LIFR, p-YAP1(S127), and YAP1 in three cancer cell lines. **b** Immunostaining for YAP1 and LIFR in three cancer cell lines. Scale bars, 10 μm. **c** Western blot analysis of LIFR and p-YAP1 (S127) protein levels in WT and cLIF cancer cells transfected with SMARTpool LIFR siRNA or control siRNA. The p-YAP1 (S127) expressions with respect to total YAP1 levels were quantified and presented as mean ± SEM ($n = 3$). At least three independent experiments were performed. **d** Western blot analysis of expression of focal adhesion molecules and SRC in WT and LIF$^{+/-}$ cancer cells transfected with YAP1 siRNA. GAPDH was used as the loading control. **e** Representative images for p-PXN (Y118) expression in WT or LIF$^{+/-}$ cancer cells transfected with YAP1 or control siRNA ($n = 3$). Alexa Fluor 488 phalloidin (green) was used to stain F-actin. Scale bars, 10 μm. **f** Representative images of the HUVEC layer replacement assay. Fixed numbers of WT or LIF$^{+/-}$ cancer cells transfected with YAP1 or control siRNA were plated onto the confluent HUVEC layer and co-cultivated for 24 h. Cancer cells were labeled with antibody against pan-cytokeratin (red) and HUVEC cells with antibody against VE-cadherin (green). Scale bars, 20 μm. **g**, **h** Quantification of displaced areas depicted in **f**. WT cancer cells (**g**). LIF$^{+/-}$ cancer cells (**h**). Invaded areas were calculated using CellSens imaging software (Olympus). Data are presented with scatter dot plot (mean ± SEM). Each black dot represents one captured image. Mann–Whitney test. **i** Immunohistochemistry for LIFR and YAP1 expression (brown) in consecutive NPC biopsy sections derived from primary or bone marrow metastatic lesions

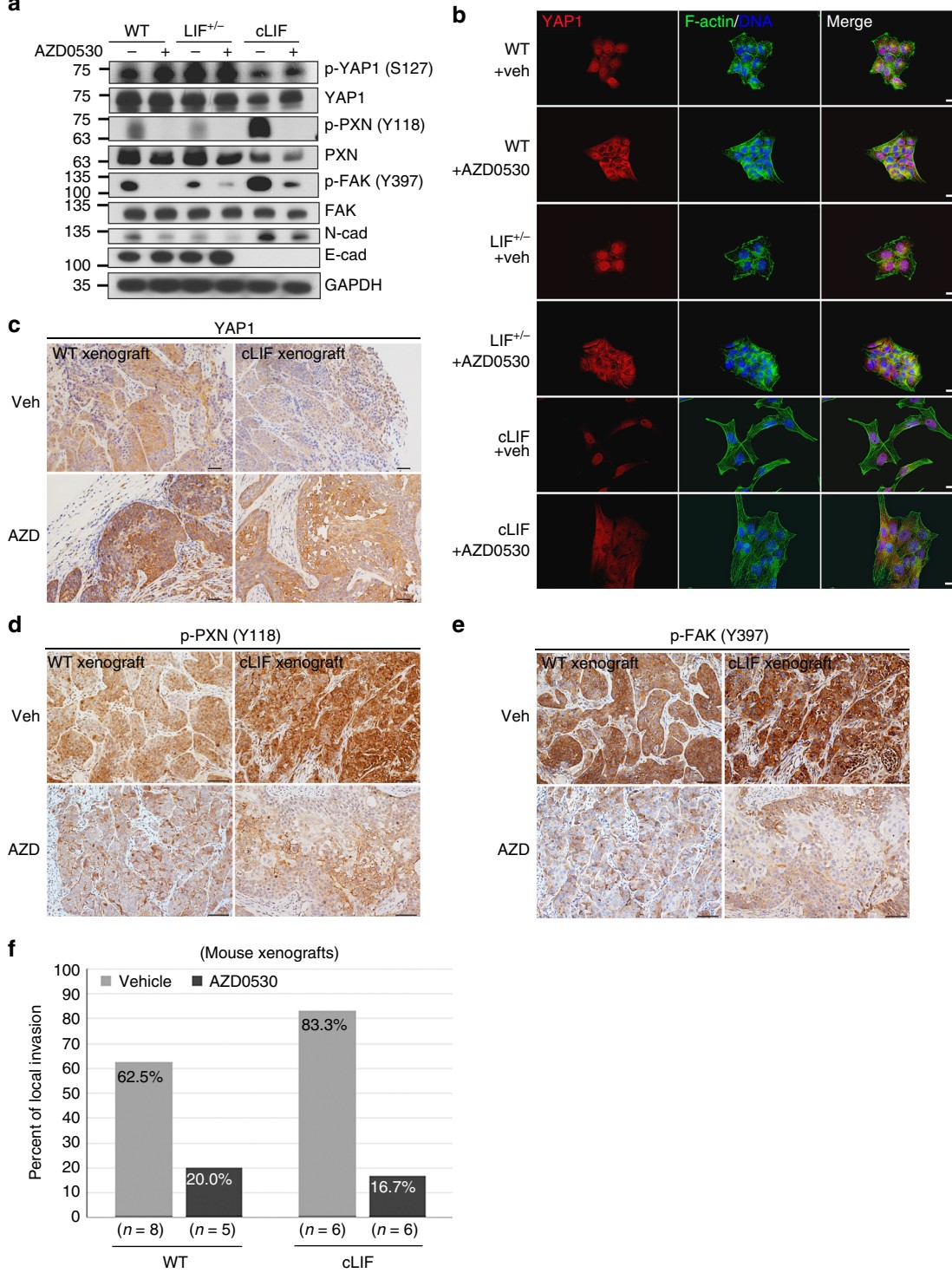

**Fig. 7** AZD0530 treatment suppresses LIF-mediated tumor invasion. **a** Western blot analysis of YAP1 and focal adhesion proteins in cancer cells treated with AZD0530 (5 μM). Protein lysates were harvested at 24 h post treatment. **b** Immunostaining for YAP1 (red) in cancer cells treated with AZD0530. Alexa Fluor 488 phalloidin (green) was used to stain F-actin. Blue, nuclear staining. Scale bars, 10 μm. **c–e** Representative images of YAP1 (**c**), p-PXN (Y118) (**d**), and p-FAK (Y397) (**e**) expression in mouse WT and cLIF xenografts treated with AZD0530 or vehicle. Scale bars, 50 μm. **f** Quantification of mouse NPC xenografts with events of local invasion based on results of hematoxylin and eosin staining. The AZD0530 treatment procedure in the mouse model is described in Methods

AZD0530 not only enhances expression of p-YAP1 (S127) but also prevents cLIF cell-induced HUVEC damage, as reflected by the increased levels of HUVEC junctional markers: CD31 and VE-cad in the AZD0530-treated groups (Fig. 8e). To further establish whether AZD0530 affects tumor vascular dissemination

in vivo, zebrafish embryonic tumor xenografts were treated with 15 μm AZD0530 at 24 h post-cancer cell injection. Vascular dissemination of tumor-like structures was measured 4 days after AZD0530 treatment. Marked reduction of disseminated tumor-like structures in zebrafish embryos treated with AZD0530 was

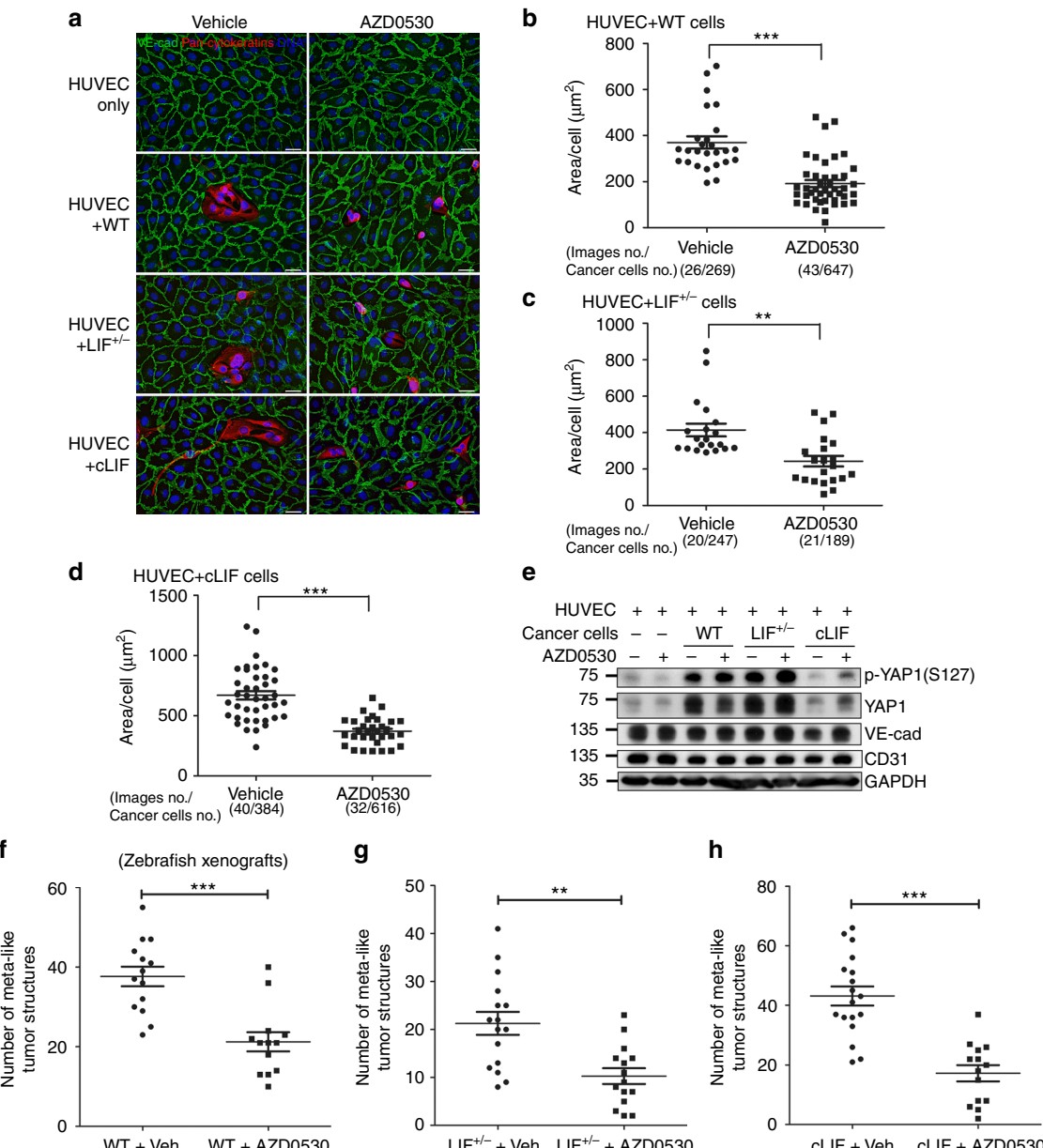

**Fig. 8** AZD0530 treatment suppresses cancer vascular dissemination. **a** HUVEC layer replacement assay. Equal numbers of cancer cells were co-cultivated with confluent HUVEC cells for 24 h in the presence of AZD0530 (5 μm) or vehicle (DMSO). Cells were fixed and labeled with antibodies against pan-cytokeratin (red) and VE-cadherin (green). Scale bars, 20 μm. Blue, nuclear staining. **b**–**d** Quantification of displaced areas described in **a**. HUVEC + WT cancer cells (**b**). HUVEC + LIF$^{+/-}$ cancer cells (**c**). HUVEC + cLIF cancer cells (**d**). Invaded areas were calculated using CellSens imaging software (Olympus). Data are presented with scatter dot plot (mean ± SEM). Each black dot represents one captured image. ** $p < 0.01$, *** $p < 0.001$, Mann–Whitney test. **e** Western blot analysis of p-YAP, VE-cadherin, and CD31 expression in co-cultivated cancer cells and HUVEC cells, as described in **a**. Total protein lysates were harvested 24 h post AZD0530 treatment using GAPDH as a loading control. **f**–**h** Quantification of disseminated tumor-like structures in zebrafish embryonic xenograft models. WT xenografts (**f**). LIF$^{+/-}$ xenografts (**g**). cLIF xenografts (**h**). The number of disseminated tumor-like structures was determined on day 4 after treatment with AZD0530 (15 μm) or vehicle (DMSO). Data are presented with scatter dot plot (mean ± SEM). Each black dot represents one fish embryo. ** $p < 0.01$, *** $p < 0.001$, Mann–Whitney test

evident [for cLIF xenografts, vehicle vs. AZD0530; mean number (SD): 43.1 (13.6) vs. 17.2 (10.2)] (Fig. 8f–h). Our data collectively suggest that AZD0530 reverses the LIF-associated cancer invasive phenotype, at least partly, by increasing cytoplasmic YAP1 and suppressing expression of the focal adhesion components p-PXN and p-FAK.

The schematic model based on our findings shows that increased cellular LIF promotes invadopodia formation, EMT process, and cancer dissemination by suppressing phosphorylation of YAP1 (S127) via LIFR, activating SRC signaling and focal adhesion assembly (Fig. 9). From the therapeutic point of view, treating LIF-high NPC cells with AZD0530 may potentially prevent cancer dissemination by inactivating YAP1-SRC-focal adhesion signaling.

## Discussion

In the present study, we focused on the molecular pathways mediating LIF-mediated NPC metastasis. Higher levels of cytoplasmic LIF and LIFR in tumors were predictive of poorer metastasis-free and recurrence-free survival of NPC patients.

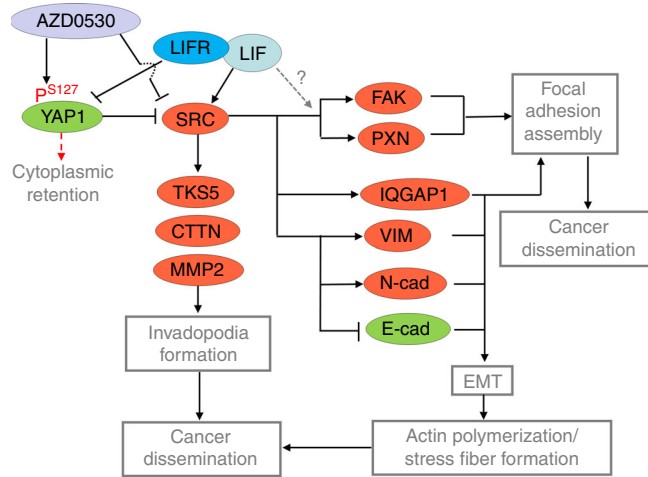

**Fig. 9** The schematic model for LIF-mediated cancer dissemination in NPC

Data obtained with our in vitro model showed that higher cellular LIF is concomitant with increased LIFR expression, consistent with the IHC results from clinical NPC samples. Accumulation of cytoplasmic LIF in NPC cells promoted EMT and invadopodia formation leading to enhanced damage of the HUVEC layer and increased vascular dissemination of tumor cells in zebrafish embryonic xenografts. In mouse tumor xenografts, higher cellular LIF was also associated with increased tumor stromal invasion. LIF-induced invasive phenotypes were accompanied by cytoskeletal reorganization, enhanced cell attachment and redistribution of focal adhesion. Mechanistically, cellular LIF appears to modulate the LIFR–YAP–focal adhesion axis to drive cancer invasion. Our results further indicate that LIFR suppresses expression of p-YAP1 (S127) in LIF-rich cancer cells. The negative correlation between reduced YAP1 expression and metastasis is supported by the observation that depletion of YAP1 in WT or LIF-low cancer cells resulted in a phenotype resembling LIF-rich cancer cells. IHC analysis of human NPC biopsies revealed significant correlations between expression of cytoplasmic LIF and LIFR as well as activated focal adhesion components. Importantly, treatment with AZD0530 impaired cLIF cancer vascular dissemination and local invasion in vitro and in vivo, and its action was linked to increased phosphorylation of YAP1 (leading to trapping of YAP1 in the cytoplasm). Our findings suggest that the LIF/LIFR–YAP1–SRC–FAK/PXN pathway serves as an ideal target to prevent NPC metastasis. While loss of YAP1 expression promotes metastasis, cytoplasmic YAP may conversely act as an active metastasis suppressor. However, the precise mechanisms require further investigation.

Recently, researchers have gained a better understanding of the multifaceted functions of LIF and LIFR in human cells. However, controversy exists regarding the roles of LIF and LIFR in human malignancies. LIF is reported to function as a growth factor in rhabdomyosarcomas[7], pancreatic carcinoma[46], choriocarcinoma[47], breast cancer[48], prostate cancer[48], and NPC[3] and as a growth inhibitory factor in cervical carcinoma[49] and medullary thyroid cancer[50]. On the other hand, dysregulation of LIFR has been described in a number of human cancers with diverse clinical relevance[3,29,51]. A recent report showed that feedback activation of LIFR–JAK1–STAT3 signaling suppresses response to HDAC inhibitors in breast cancer[52], suggesting a role of LIFR in therapeutic resistance. Furthermore, LIFR functions as a potential metastasis suppressor in human breast cancer cells through activation of the Hippo-YAP pathway[29]. Hippo-YAP signaling modulates diverse functions, and precise control of the location and level of YAP/TAZ is crucial for accurate regulation.

Recently, YAP was implicated as both an oncogene that stimulates tumor growth and a tumor suppressor gene promoting apoptosis[22,23,25,53]. We demonstrated that p-YAP1 (S127) was markedly downregulated in LIF-rich cells (cLIF) and siRNA knockdown of LIFR in cLIF cells enhanced YAP1(S127) phosphorylation, implying that intracellular LIFR signaling acts as an upstream regulator of YAP1 and LIFR functions as an oncogene promoting NPC progression. The conflicting results regarding the role of LIFR–YAP1 in human malignancies may be attributed to the diverse properties of different cancers and stages of tumor progression. Utilizing the Oncomine platform[54] and cBioPortal web resource[40] to analyze LIFR expression and genetic alterations in cancer genomics datasets, we observed that LIFR is overexpressed in melanomas, CNS tumors, lymphomas, and prostate cancers, but downregulated in breast, colorectal, and lung cancers. One of the potential causes leading to the diverse LIFR expression in human malignancies is the presence of agonist or antagonist in tumor microenvironment. Chronic or repeated exposure of cancer cells with environmental LIF can lead to downregulation of LIFR. Several soluble receptors for IL-6 family cytokines have been identified, including sLIFR, sIL-6R, sCNTFR, and sIL-11R[55,56]. Secreted LIFR or soluble LIFR is recognized as a natural antagonist of LIF[55,57–59]. It competes the binding of extracellular LIF to membrane-bound LIFR and thus attenuates the exogenous LIF-mediated signaling and biological effects. The genetic alterations of LIFR could also attribute to its diverse roles in cancer biology. Mutations of LIFR are reported to affect protein functions[60–62]. Multiple genetic alterations of LIFR have been detected in breast, lung, liver, melanoma, prostate and head-and-neck cancers (data extracted from cBioportal database). Unpublished NGS data from nearly 60 NPC patient tumor samples also revealed genetic alterations of LIFR in NPC tumors. However, the issue of whether those genetic changes affect the functions of LIFR and associated mechanisms requires further investigation.

LIF has been reported to exist as a secreted cytokine or be retained in the intracellular compartments[34–36]. Our sequencing data of 157 NPC biopsy samples identified multiple single nucleotide variations across the LIF coding region, including two missense mutations. By comparing with available cancer mutation data extracted from cBioportal database, we further observed other LIF mutations residing within the signal peptide region in the melanoma studies. Melanomas have been reported to express high level of LIF and LIFR, which are associated with poorer prognosis[11,13,31]. Considering the similar alterations of LIF/LIFR between NPC and melanoma, systemically analyzing the omics data of these two cancers will provide insightful information regarding to the role of LIF/LIFR in cancers.

Improved understanding of the roles of LIF in cancer metastasis should facilitate the prevention of cancer dissemination through sequestering of its regulators or downstream effectors. In this study, we have unveiled a novel role of AZD0530 as a potent YAP1 and metastasis inhibitor. From a clinical perspective, AZD0530 treatment presents an ideal strategy to disrupt the metastatic process in cancers associated with LIF, LIFR, or YAP1 activation.

## Methods

**Characteristics of patients involved in survival analysis**. All patients with no history of radiotherapy treatment or metastasis completed treatment within 10 weeks (dose ≥ 6600 cGy). The general characteristics of the LIF study participants ($n = 185$) are summarized in Supplementary Table 1 and baseline characteristics of the LIFR study population ($n = 150$) in Supplementary Table 3. All individuals were followed up for more than 3 years after treatment. NPC tumors were histologically confirmed by pathologists. This study was approved by the Institutional Review Board of Chang-Gung Memorial Hospital (IRB104-9216B), Taiwan. Written informed consent was received from participants prior to inclusion in the study.

**Establishment of LIF mutant clones using TALEN technology**. To generate cytoplasmic LIF-rich cancer (cLIF) and heterozygous LIF knockout (LIF$^{+/-}$) strains, we conducted LIF gene editing with site-specific transcription activator-like effector nuclease (TALEN) technology. The first exon and immediate flanking regions of human LIF were scanned for putative TALEN binding pairs using the TAL Effector Nucleotide Targeter 2.0[63] (Supplementary Figure 2). The sequences were cloned into Talen vector with the puromycin resistance gene. BM1 cells were seeded in a 6-well plate at a density of $4 \times 10^4$ cells per well for one day and a pair of TALEN constructs for LIF gene editing transfected into cells using Fugene HD transfection reagents (Promega). The day after transfection, 3 μg/ml puromycin was administered for 3 days. Individual clones were isolated and sites of editing verified via PCR and Sanger sequencing analyses. Among ~30 isolated cell clones, we selected one with alterations in five nucleotides within the signal peptide region of LIF (cLIF clone) and one with loss of the initiating codon in one allele (LIF$^{+/-}$ clone) for further investigation. We were unable to successfully culture the LIF knockout clone (LIF$^{-/-}$) due to unhealthy cell conditions and difficulties in cell propagation.

**HUVEC layer displacement assay**. The invasive ability of cancer cells was evaluated using the HUVEC layer displacement assay. HUVEC cells were seeded into collagen-coated chamber slides or coverslips at a density of $5 \times 10^5$ cells/ml and grown to confluence. A fixed number of cancer cells ($7 \times 10^3$ cells) was added on top of the HUVEC layer and co-cultivated for 24 h. Cells were fixed and co-stained with antibodies against carcinoma markers (pan-cytokeratin) and endothelial markers (VE-cadherin or CD31) to define the damaged areas of the HUVEC layer. Disrupted areas were calculated using CellSens imaging software (Olympus).

**Western blotting**. Cells were harvested in NP-40 lysis buffer containing a protease inhibitor cocktail (Roche). Equal amounts of protein were separated via sodium dodecyl sulfate-polyacrylamide gel electrophoresis and transferred to poly-vinylidene fluoride membranes. Blots were probed with specific primary antibodies against LIF (Abcam, ab135629, 1:500), LIFR (Santa Cruz Biotechnology, sc-515337, 1:500), phospho-YAP1 (Cell Signaling, 13008, 1:2000), YAP1 (Cell Signaling, 14074, 1:2000), phospho-FAK (Invitrogen, 700255, 1:5000), FAK (Santa Cruz Biotechnology, sc-558, 1:500), PECAM-1 (CD31) (Santa Cruz Biotechnology, sc-376764, 1:500), VE-cad (Santa Cruz Biotechnology, sc-9989, 1:1000), E-cad (BD Transduction Laboratories, 610181, 1:5000), phospho-PXN (Y118) (Abcam, ab109547, 1:4000), PXN (BD Transduction Laboratories, 610052, 1:10,000), phospho-p70S6K1 (T389) (Cell Signaling, 9234, 1:2000), p70S6K1 (Abcam, ab32529, 1:10,000), N-cad (Abcam, ab76011, 1:10,000), VIM (Sigma-Aldrich, V5255, 1:500), IQGAP1 (Santa Cruz Biotechnology, sc-81906, 1:1000), phospho-SRC (Cell Signaling, 6942, 1:2000), SRC (Cell Signaling, 2109, 1:2000), TKS5 (Proteintech, 18976-1-AP, 1:1500), CTTN (Abcam, ab81208, 1:10,000), MMP2 (Proteintech, 10373-2-AP, 1:800), and GAPDH (Abcam, ab8245, 1:5000) by incubation with horseradish peroxidase-conjugated secondary antibody and developed with enhanced chemiluminescence detection reagent (GE Healthcare). Uncropped scans of all immunoblots are provided in Supplementary Figure 6-7.

**Immunohistochemistry**. Immunohistochemistry was conducted using the Leica BOND-MAX automated system and Bond Polymer Refine Detection Kit (Leica Microsystems, DS9800) according to the manufacturer's protocol. Briefly, tissue sections were deparaffinized with Bone Dewax Solution and treated with the Epitope Retrieval Solution 1 (Citrate buffer) or Solution 2 (EDTA-buffer pH 8.8) at 98 °C for 20 min. After washing, 3–4% (v/v) hydrogen peroxide was added to block endogenous peroxidase activity. Tissues were washed and incubated with primary antibodies for 30 min followed by polymer for 15 min and developed with 3,3-diaminobenzidine (DAB) for 10 min. For the negative control, the primary antibody was omitted and replaced with blocking buffer containing the same amount of IgG from non-immune rabbit or mouse serum. The staining score was defined according to staining intensity and extent ("−", "+", "++" or "+++" and ratio of the positive cancer cell number relative to total cancer cells, respectively). The majority of NPC biopsies were pre-treatment primary tumors derived from NPC patients diagnosed with complete tumor remission, tumor relapse, or metastasis after treatment. Only those samples grouped in the metastatic lesions are tumor biopsies derived from metastatic sites. IHC results were reviewed by pathologists.

**Physical measurement of 3D-gelatin invasion assays**. A total of $3 \times 10^4$ cells were plated on the surface of 35 mm culture dishes coated with stiff gelatin matrix (6% gelatin dissolved in PBS) for the measurement of invasion ability. The height of matrix is approximately 160 μm and the stiffness of gelatin matrix for 6% gelatin is about 30 kilopascal (kPa). The invasion depth and distance of cancer cells were continuously measured by time-lapse phase contrast vertical scanning (Olympus IX83).

**Mouse xenograft tumor models**. To investigate local tumor invasion and drug response, we used non-obese diabetic (NOD)/severe-combined immunodeficient (SCID) mice from BioLASCO (Taiwan) to establish mouse tumor xenografts by subcutaneously injecting cancer cells ($2 \times 10^6$ cells in 100 μl PBS) expressing luciferase 2 into the legs of 7 week-old male NOD/SCID mice. For the drug response

experiments, AZD0530 (LC Laboratories) was dissolved in DMSO to generate a concentration of 200 mg/ml and diluted in 0.2% Tween 80 /PBS before administration. When tumors reached a size of ~50 mm$^3$, mice were treated with either vehicle (DMSO in 0.2% Tween 80/PBS) or 30 mg/kg AZD0530 by oral gavage once a day for 6 consecutive days with one day of rest for a period of 4 weeks. Tumor progression was monitored once a week with an in vivo imaging system (IVIS). Briefly, mice were anesthetized with isoflurane gas and injected intraperitoneally (150 mg/kg) with D-luciferin solution (Promega) and bioluminescent image measurements obtained using an IVIS Spectrum (Xenogen IVIS 100; Caliper). Mice were maintained under specific pathogen-free conditions at the Laboratory Animal Center of Chang-Gung University (Taoyuan, Taiwan). All animal experiments were handled according to the accepted principles of laboratory animal care and approved by the animal committee of Chang-Gung University and National Central University (Taoyuan, Taiwan).

**Zebrafish embryonic xenograft tumor model**. To analyze vascular dissemination, we established zebrafish embryonic tumor xenograft models by injecting 250 cancer cells expressing LifeAct-RFP into the yolk sac of 48 h post-fertilization (hpf) embryos of Tg(fli1a:EGFP)y1, an EGFP-positive endothelial transgenic zebrafish line, and measured the vascular dissemination of tumor-like structures at 6 days post-injection. To evaluate the cancer drug response, we treated zebrafish embryos with 15 μm AZD0530 or vehicle (DMSO) at 24 h post-injection with cancer cells and measured vascular dissemination of tumor-like structures after 4 days. Growth and dissemination of tumor-like structures in living fish embryos were visualized under a high-resolution fluorescent microscope (Olympus IX83). At the end of the experimental period, fish embryos were fixed with 4% paraformaldehyde for 16 h at 4 °C. Vascular dissemination of tumor-like structures (tumor area ≥ 10 μm$^2$) was determined using CellSens imaging software (Olympus). The zebrafish experiments were conducted according to the guidelines of the Institutional Animal Care and Use Committee (IACUC) of National Health Research Institutes (NHRI), Taiwan.

**Real-time monitoring of cellular responses**. The xCelligence real-time analyzer (ACEA Biosciences) was utilized to monitor the invasive ability of cancer cells through measuring altered cell impedance of the endothelial layer. Prior to cell seeding, the background impedance of each E-plate was determined using plates loaded with 60 μl culture medium per well. HUVEC cells ($1 \times 10^4$ cells in 100 μl/ well) were plated and allowed to grow until confluence at 37 °C, prior to the addition of cancer cells. Cell impedance was recorded every 10 min for 4 days. All experiments were performed in triplicate.

**Statistics**. Statistical analyses were performed using GraphPad Prism 5 (GraphPad Software) or SPSS 16.0 (SPSS). Kaplan–Meier survival and log-rank tests were applied to compare survival times between groups. The multivariate Cox proportional hazards model was used to identify independent predictors for metastasis-free and recurrence-free survival. IHC analyses of immunoreactivity in human NPC biopsies were compared using the $\chi^2$ test. Spearman's rank correlation coefficient was applied to evaluate the correlations between IHC results. Microscopic quantification of damaged areas in HUVEC displacement assays and measurement of tumor-like structures in zebrafish embryonic xenografts were performed using CellSens imaging software (Olympus). The Mann–Whitney test was used to evaluate the differences among the in vivo experimental groups. All statistical tests were two-sided and $p$ values < 0.05 considered statistically significant.

## Data availability
The authors declare that all data supporting the findings of this study are available within this published article and its Supplementary Information files and from the corresponding author upon reasonable request.

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

## Acknowledgements

This work was funded by the Ministry of Science and Technology, Taiwan (MOST 104-2321-B-008-003-MY3, MOST 105-2314-B-182-032-), a grant from the VGHUST Joint Research Program (VGHUST106-G4-2-1), and a grant from Chang-Gung Memorial Hospital (CMRPG3C1922). We appreciate the bioinformatic service provided by the National Core Facility for Biopharmaceuticals (MOST 107-2319-B-400-001). We thank Lin. I, Wei-Yen Woon, Yi-Teng Hsiao, Hsiang-Ying Chen, Chun-Yu Liu (Department of Physics, National Central University, Taiwan), and members of the Cancer Center (Linkou Chang-Gung Memorial Hospital) for their invaluable help.

## Author contributions

S.-C.L.: conception, design, methodology, most of the experiments, writing, funding acquisition; T.H.: supervision, resources, review, funding acquisition; N.-M.T.: acquisition of clinical IHC sections, analysis and interpretation of clinical data, funding acquisition; Y.-S.C.: resources, material support; A.-K.C.: sequencing, mutation analysis; S.S.J.: computational analysis; C.-N.O.: animal experiments; C.-H.Y.: zebrafish experiments; C.H.: interpretation of IHC data; Y.-P.L.: immunohistochemistry.

## Additional information

**Competing interests:** The authors declare no competing interests.

