## [Peer Review File · Nature Communications]

Reviewers' Comments:

Reviewer #1:

Remarks to the Author:

This is a very interesting report showing that in contrast to the breast cancer signaling scenario, where Leukemia Inhibitory Factor Receptor (LIFR) is known to act as a tumor suppressor, the enhanced signaling by Leukemia Inhibitory Factor (LIF) and LIF(Receptor)R in nasopharyngeal carcinoma (NPG) promotes metastatic potential.

The report is clear and it is well presented. The quality of the data is very good. There are many important observations described in the manuscript in cell culture models and xenografts. However, a clear mechanism that one could present in a "graphic abstract" at the end, for the Readers of "Nature Communications" so they could grasp it quickly and take it to their laboratory benches to verify it, is not that clear.

The following general changes are suggested to improve the manuscript:

1. The nature of increased intra-cellular LIF needs more analysis. If a mutation in the signal peptide is found in NPG that would be a "foundling stone" of the report.
2. The rationale for the use of Src and Abl inhibitor is not clear. Indeed, YAP was shown to be Yes- and Src-kinase associated protein but that applies to intestinal cells and is really context/cell dependent. The protein complex of Src and Yes with YAP is weak, but enhanced with accessory proteins. There are examples of YAP being phosphorylated Src and Abl kinases, but this is again subtle and context dependent.
3. A longer discussion of why the NPG is so different from breast cancer would be required. The study from the laboratory of Dr. Li Ma, in Texas USA, has showed nicely the mechanism of tumor suppression by LIFR. The differences between the mechanisms deserve more detailed discussion.

Reviewer #2:

Remarks to the Author:

This manuscript investigates the function of cytoplasmic LIF, and its cognate receptor LIFR, in regard to the invasive phenotypes of nasopharyngeal carcinoma (NPC) cells. The authors find that cytoplasmic LIF promotes dissemination of carcinoma cells through a mesenchymal, single-cell mode of invasion that is associated with features of the EMT program and the appearance of invadopodia. It is argued that this transition is mediated through a signaling axis that involves YAP and the focal adhesion kinases FAK and PXN. The study involves the use of numerous relevant experimental systems and as well as analysis of clinical NPC samples. The experiments, in most cases, seem to well performed and their findings may hold interesting implications for those studying cancer cell invasion, LIF/LIFR signaling and YAP biology. However a number of outstanding issues temper the enthusiasm of this reviewer and need to be resolved.

Perhaps most importantly, independent of all other considerations, is the rather unusual, if not strange foundation of this paper. To quote from the Abstract: "Cytoplasmic LIF reprograms the invasive mode from collective to mesenchymal migration ..." . Indeed, much of the structure of the present manuscript is based on this conclusion. The authors refer to an earlier paper of theirs that appeared in the J. of Clin. Investigation, in which no mention is made of "cytoplasmic LIF", and the authors do not seem to be concerned about the fact that cytokines like LIF do not operate in the cytoplasm but instead in the extracellular space. How can they conclude that "cytoplasmic LIF" is indeed the agent of change when there seems to be no precedent in their own work or in the literature (this reviewer has attempted to find earlier references in PubMed about "cytoplasmic LIF") about this protein? Hence, while secreted, extracellular LIF has extensive documentation in

the literature whereas a cytoplasmic counterpart does not. Given all this, how can the entire edifice of this paper be upheld on such a shaky foundation? It is difficult to see how this manuscript can be published given this and other reservations expressed below.

Some additional comments:

1. This group has previously published on the correlation between LIF expression and disease prognosis in NPC patients. Is the data presented here, specifically in Figure 1, different in that only cytoplasmic LIF is being scored? Or is the scoring of LIF essentially the same and this patient group represents further validation and an extension to metastatic samples? How does the staining in Figure 1A of NPC recurrence and NPC metastasis compare to an NPC primary tumor?
2. The use of TALENs to mutate the signal peptide region of LIF is an interesting approach to study the cytoplasmic function of LIF. But I remain concerned that the cytoplasmic staining observed in patient samples could simply represent a surrogate marker of cells that produce and secrete abundant levels of LIF. This would seem to be consistent with the prognostic significance of LIF that is reported here. To put it another way, are the engineered LIFc clones faithful representatives of the NPC carcinoma cells observed in Figure 1? Given that there is no evidence for this type of signal peptide mutation in LIF, the LIFc clones could be rather idiosyncratic.
3. There are a number of problems with the conclusions drawn at the end of the section titled "Characterization of established LIF mutant cancer clones." For one, it states that the LIFc cells are not desensitized by exogenous LIF. It would be better, and more accurate, to simply state that they are not responsive to exogenous LIF. Second, it states that "these findings clearly indicate that cytoplasmic LIF and LIFR are co-expressed," which doesn't seem like the right conclusion here. Do the authors mean that cLIF can regulate the expression of LIFR? Or that the cLIF is functional in the engineered clones?
4. In regard to the vascular invasion assays where LIFc cells are able to disrupt the HUVEC layer, are the authors sure that no LIF is secreted from the LIFc cells? In general this seems like an important control to include, but especially so here in order to ensure that secreted LIF is not acting directly on the endothelial cells. Also, do LIFc cells exhibit enhanced migration in a transendothelial migration assay?
5. The western blots are confusing in these co-culture assays. Do these lysates represent bulk lysates of HUVEC and BM1 cells?
6. Using the zebrafish xenograft model the authors state the LIF is critical for "cancer vascular evasion." To clarify, in this model does this represent intravasation or extravasation?
7. The authors argue that LIFc switches the mode of invasion from collective to mesenchymal (single-cell). The evidence for this comes largely from a wound-healing assay. Can a similar behavior be observed using 3D assays that have been used to study this behavior (see Cheung, *Cell*, 2014; Ye, *Nature*, 2015)?
8. The weakest aspect of this study involves the connection between LIF, LIFR and the YAP pathway. There are a multitude of issues here that make interpretation of the results difficult. 1. LIFc/LIFR-high cells have low YAP1 levels but knockdown of LIFR leads to increased phosphorylation of YAP. These two scenarios (p-YAP vs. total YAP) are not equivalent as the authors seemingly argue. 2. There is also the suggestion that LIFc cells have lower p-YAP but relative to total YAP this doesn't seem to be true. 3. The changes in p-PXN and p-FAK after YAP knockdown aren't very convincing. Thus, with the data presented it is very difficult to support the mechanistic model they have put forth where LIFc inactivates YAP. What happens if a constitutively active YAP is expressed in LIFc cells? And does their model support the idea that YAP plays some sort of functional role in cells with low LIF/LIFR?
9. The authors describe a correlation between LIFR expression and cytoplasmic YAP in clinical NPC samples (Fig. 6c) but in the LIFc cells YAP seems to exhibit a nuclear localization (Fig. 7b). How do the authors reconcile this discrepancy?
10. The use of AZD0530, although it has some effects, seems somewhat arbitrary. Do LIFc cells have higher levels of active Src? Is Src downstream/upstream of LIFc/LIFR or does it function in a parallel pathway. Does AZD0530 have any effect on the viability/proliferation of the LIFc cells? How does this compound affect the LIFwt and LIF+/- cells? Some basic controls are needed here.

11. The mechanism put forth by the authors is rather complicated and it seems a simpler interpretation could explain their findings. In Figure 2 and 3 there is pretty clear evidence that the LIFc cells have undergone an EMT. The EMT program has been shown to regulate tumor cell invasion (Nieto, Cell, 2016), invadopodia formation (Eckert, Cancer Cell, 2011), transendothelial cell migration (Drake, MBoC, 2009), and the expression of focal adhesion molecules (Shibue, Cancer Cell, 2013). Thus, it seems that many, if not all, of the observations reported here can be explained by the fact that LIFc cells are more mesenchymal than the LIFwt and LIF+/- cells.

12. References 12 and 19 are the same.

Point-by-point responses to reviewers' comments

Reviewer #1:

This is a very interesting report showing that in contrast to the breast cancer signaling scenario, where Leukemia Inhibitory Factor Receptor (LIFR) is known to act as a tumor suppressor, the enhanced signaling by Leukemia Inhibitory Factor (LIF) and LIF(Receptor)R in nasopharyngeal carcinoma (NPC) promotes metastatic potential.

The report is clear and it is well presented. The quality of the data is very good. There are many important observations described in the manuscript in cell culture models and xenografts. However, a clear mechanism that one could present in a “graphic abstract” at the end, for the Readers of “Nature Communications” so they could grasp it quickly and take it to their laboratory benches to verify it, is not that clear.

Our Response:

Thank you for this very constructive comment. Based on our findings, we propose a graphic abstract in this revised version. It is presented on page 24 and Figure 9. The schematic model shows that increased cellular LIF promotes invadopodia formation, EMT process, and cancer dissemination by suppressing phosphorylation of YAP1 (S127) via LIFR, activating SRC signaling and focal adhesion assembly. Treating LIF-high NPC cells with AZD0530 suppresses cancer dissemination by inactivating LIF downstream YAP1-SRC-focal adhesion signaling.

Fig. 9 The schematic model for LIF-mediated cancer dissemination in NPC.

The following general changes are suggested to improve the manuscript:

1. The nature of increased intra-cellular LIF needs more analysis. If a mutation in the signal peptide is found in NPG that would be a “founding stone” of the report.

Our Response:

We have followed the suggestion to analyze the genetic alterations of LIF gene in 157 clinical NPC biopsy samples. Results of sequencing data identified a variety of LIF single nucleotide variations (SNVs), including two missense mutations with one residing within the signal peptide region (G20L) (page 9, Fig. 1k). The LIF and LIFR expression is higher in the patient harboring LIF signal peptide mutation (page 10, Fig. 1l), suggesting that LIF signal peptide mutation may upregulate intracellular LIF and LIFR expression. We also analyzed the LIF mutation spectrum of large-scale cancer genomics data extracted from the cBioPortal database^{1,2} and found other LIF signal peptide mutations in melanoma studies (page 10, Supplementary Fig. S1, Supplementary Table S3). Whether other SNVs or mutations are associated with LIF/LIFR expression will need further experimental and clinical validations.

1k

1l

Fig.1k Single nucleotide variations in NPC biopsy samples (n = 157). Circles are colored with respect to the corresponding mutation types.

Fig.1l LIF and LIFR expression in tumors from NPC patient carrying LIF signal peptide mutation (G20L).

Supplementary Figure S1

Supplementary Fig. S1 Spectrum of LIF genetic alterations in cancers. **a** Summary of cross-cancer genetic alterations for LIF gene. **b** Diagram of LIF mutations. Circles are colored with respect to the corresponding mutation types. Data were extracted from cBioPortal database. A total of 216 studies were included in the analysis.

2. The rationale for the use of Src and Abl inhibitor is not clear. Indeed, YAP was shown to be Yes- and Src-kinase associated protein but that applies to intestinal cells and is really context/cell dependent. The protein complex of Src and Yes with YAP is weak, but enhanced with accessory proteins. There are examples of YAP being phosphorylated Src and Abl kinases, but this is again subtle and context dependent.

Our Response:

Thanks for the suggestion, we have revised our description about the use of AZD0530 in this study (page 21, line 8 to line 15). Further, we also conducted

survival assays to evaluate the toxicity of each cell lines in response to AZD0530 treatment (page 21-22, Supplementary Fig. S5). Our data demonstrated that the LIFc cells were more resistant to AZD0530 treatment ($IC_{50} = 1.73 \times 10^{-5}$ M) whereas the LIF^{+/-} cells were more sensitive to AZD0530 treatment compared with LIF wild-type cells ($IC_{50} = 8.93 \times 10^{-7}$ M).

Supplementary Figure S5

Supplementary Fig. S5 Assessment of the IC_{50} values of AZD0530 for WT and LIF mutant cancer cells. **a, c, e** Real-time measurement of cell survival in response to various doses of AZD0530 treatment. Values of cell survival were normalized with respect to the time of AZD0530 addition. Values were presented as means and SD of quadruplicate experiments. **b, d, f** The AZD0530-induced cell toxicity was determined by calculating the IC_{50} after treatment. Normalized cell index was displayed against the logarithm of concentration. The values of IC_{50} were calculated using the RTCA software (ACEA Biosciences).

3. A longer discussion of why the NPG is so different from breast cancer would be required. The study from the laboratory of Dr. Li Ma, in Texas USA, has showed nicely the mechanism of tumor suppression by LIFR. The differences between the mechanisms deserve more detailed discussion.

Our response:

Thanks for the very constructive suggestion. We have added extensive discussions about the different roles of LIFR-YAP1 between breast cancer and NPC (page 26-29). We also utilized genome-wide data from the OncoPrint platform³ and cBioPortal web resource⁴ to provide information that LIFR may play distinct roles in different types of cancers.

Issues regarding the distinct roles of LIFR in cancer were discussed as following: Dysregulation of LIFR has been described in a number of human cancers with diverse clinical relevance⁵⁻⁸. A recent report showed that feedback activation of LIFR-JAK1-STAT3 signaling suppresses response to HDAC inhibitors in breast cancer⁹, suggesting a role of LIFR in therapeutic resistance. Furthermore, LIFR functions as a potential metastasis suppressor in human breast cancer cells through activation of the Hippo-YAP pathway⁷. Hippo-YAP signaling modulates diverse functions, and precise control of the location and level of YAP/TAZ is crucial for accurate regulation. Recently, YAP was implicated as both an oncogene that stimulates tumor growth and a tumor suppressor gene promoting apoptosis¹⁰⁻¹³. We demonstrated that p-YAP1 (S127) was dramatically downregulated in LIF-rich cells (LIFc) and siRNA knockdown of LIFR in LIFc cells enhanced YAP1(S127) phosphorylation, implying that intracellular LIFR signaling acts as an upstream regulator of YAP1 and LIFR functions as an oncogene promoting NPC progression. The conflicting results regarding the role of LIFR-YAP1 in human malignancies may be attributed to the diverse properties of different cancers and stages of tumor progression. Utilizing the OncoPrint platform³ and cBioPortal web resource⁴ to analyze LIFR expression and genetic alterations in cancer genomics datasets, we observed that LIFR is overexpressed in melanomas, CNS tumors (glioblastoma multiforme, glioma, anaplastic astrocytoma), lymphomas (follicular lymphoma, diffuse large B-cell lymphoma) and prostate cancers, but downregulated in breast, colorectal, and lung cancers. One of the potential causes leading to the diverse LIFR expression in human malignancies is the presence of agonist or antagonist in tumor microenvironment. Chronic or repeated exposure of cancer cells with environmental LIF can lead to downregulation of LIFR. Several soluble receptors for IL-6 family cytokines have been identified, including sLIFR, sIL-6R, sCNTFR, and sIL-11R^{14,15}. Secreted LIFR or soluble LIFR is recognized as a natural antagonist of LIF¹⁶⁻¹⁹. It competes the

binding of extracellular LIF to membrane-bound LIFR and thus attenuates the exogenous LIF-mediated signaling and biological effects. In addition to the environmental factors, the genetic alterations of LIFR could attribute to its diverse roles in cancer biology. Mutations of LIFR are reported to affect protein functions²⁰⁻²². Multiple genetic alterations of LIFR have been detected in breast, lung, liver, melanoma, prostate and head-and-neck cancers (data extracted from cBioportal database). Unpublished NGS data from nearly 60 NPC patient tumor samples also revealed genetic alterations of LIFR in NPC tumors. However, the issue of whether those genetic changes affect the functions of LIFR and associated mechanisms requires further investigation.

LIF has been reported to exist as a secreted cytokine or be retained in the intracellular compartments²³⁻²⁵. Our sequencing data of 157 NPC biopsy samples identified multiple single nucleotide variations across the LIF coding region, including two missense mutations. By comparing with available cancer mutation data extracted from cBioportal database, we further observed other LIF mutations residing within the signal peptide region in the melanoma studies. Melanomas have been reported to express high level of LIF and LIFR, which are associated with poorer prognosis²⁶⁻²⁸. Considering the similar alterations of LIF/LIFR between NPC and melanoma, systemically analyzing the omics data of these two cancers will provide insightful information regarding to the role of LIF/LIFR in cancers.

Reviewer #2

This manuscript investigates the function of cytoplasmic LIF, and its cognate receptor LIFR, in regard to the invasive phenotypes of nasopharyngeal carcinoma (NPC) cells. The authors find that cytoplasmic LIF promotes dissemination of carcinoma cells through a mesenchymal, single-cell mode of invasion that is associated with features of the EMT program and the appearance of invadopodia. It is argued that this transition is mediated through a signaling axis that involves YAP and the focal adhesion kinases FAK and PXN. The study involves the use of numerous relevant experimental systems and as well as analysis of clinical NPC samples. The experiments, in most cases, seem to well performed and their findings may hold interesting implications for those studying cancer cell invasion, LIF/LIFR signaling and YAP biology. However a number of outstanding issues temper the enthusiasm of this reviewer and need to be resolved.

Perhaps most importantly, independent of all other considerations, is the rather unusual, if not strange foundation of this paper. To quote from the Abstract: "Cytoplasmic LIF reprograms the invasive mode from collective to mesenchymal migration ..." . Indeed, much of the structure of the present manuscript is based on this conclusion. The authors refer to an earlier paper of theirs that appeared in the *J. of Clin. Investigation*, in which no mention is made of "cytoplasmic LIF", and the authors do not seem to be concerned about the fact that cytokines like LIF do not operate in the cytoplasm but instead in the extracellular space. How can they conclude that "cytoplasmic LIF" is indeed the agent of change when there seems to be no precedent in their own work or in the literature (this reviewer has attempted to find earlier references in PubMed about "cytoplasmic LIF") about this protein? Hence, while secreted, extracellular LIF has extensive documentation in the literature whereas a cytoplasmic counterpart does not. Given all this, how can the entire edifice of this paper be upheld on such a shaky foundation? It is difficult to see how this manuscript can be published given this and other reservations expressed below.

Our response:

Thanks for the reviewer's constructive comments. We have provided more evidence to show that cytoplasmic /intracellular LIF could have biological functions in NPC progression. LIF has been reported to exist as a secreted cytokine or be retained in the intracellular compartments²³⁻²⁵. Further, signal peptide mutations have been reported to result in accumulation of cytoplasmic cytokines, leading to receptor self-association and increased constitutive signal transduction^{29,30}. In NPC, elevated

cellular LIF expression correlated with LIFR expression and both are markedly correlated with poorer metastasis-free survival. We showed that higher intracellular LIF conferred cells resistant to the uptake of environmental LIF and therefore sustained the expressions of LIFR and downstream activated p70S6K1 (page 11-12, Fig. 2f, g). In addition, we analyzed the genetic alterations of LIF gene in clinical NPC biopsy samples. Results of sequencing data identified a variety of LIF single nucleotide variations (SNVs), including two missense mutations with one residing within the signal peptide region (G20L) (page 9, Fig. 1k). The LIF and LIFR expression is higher in the patient harboring LIF signal peptide mutation (page 10, Fig. 1l). We also analyzed the LIF mutation spectrum in large-scale cancer genomics data extracted from the cBioPortal database ^{1,2} and found other LIF signal peptide mutations in melanoma studies (page 10, Supplementary Fig. S1, Supplementary Table S3). Whether other SNVs or mutations are associated with LIF/LIFR expression will need further experimental and clinical validations.

Fig. 2f, g

Fig. 2f Live images of LIF uptake in cancer cells expressing LifeAct-RFP. Recombinant LIF proteins were pre-labeled with ATTO 488 green fluorescent dye. Images were captured 40 minutes post LIF addition.

Fig. 2g Time-course analysis of LIFR expression and p70S6K1 activation in LIF (30 ng/ml)-stimulated cells using GAPDH as a loading control.

Figure 1k, l

1k

1l

Fig.1k Single nucleotide variations in NPC biopsy samples (n = 157). Circles are colored with respect to the corresponding mutation types.

Fig.1l LIF and LIFR expression in tumors from NPC patient carrying LIF signal peptide mutation (G20L).

Supplementary Figure S1

Supplementary Fig. S1 Spectrum of LIF genetic alterations in cancers. **a** Summary of cross-cancer genetic alterations for LIF gene. **b** Diagram of LIF mutations. Circles are colored with respect to the corresponding mutation types. Data were extracted from cBioPortal database. A total of 216 studies were included in the analysis.

Some additional comments:

1. This group has previously published on the correlation between LIF expression and disease prognosis in NPC patients. Is the data presented here, specifically in Figure 1, different in that only cytoplasmic LIF is being scored? Or is the scoring of LIF essentially the same and this patient group represents further validation and an extension to metastatic samples? How does the staining in Figure 1A of NPC recurrence and NPC metastasis compare to an NPC primary tumor?

Our response:

- (1) All clinical samples in this study belong to an independent sample set from our previous study. All cases without previous radiotherapy treatment nor metastasis complete the treatment time within 10 weeks (Doses $\geq 6,600$ cGy). One of the reasons to explain why the previous paper did not correlate LIF expression with metastasis is that we started the study of positron emission tomography (PET) in the beginning of 2004. In our previous study, over half of the cases recruited did not have PET information. In this study, only 16.8% cases did not have PET examination before the radiotherapy treatment, that is different from the previous paper.
- (2) In this study, only the cytoplasmic LIF immunoreactivity was scored. The staining score was defined according to staining intensity and extent (“-”, “+”, “++” or “+++” and ratio of the positive percentage of cancer cells relative to total cancer cells, respectively) as described in the section of Methods (page 33).
- (3) We analyzed the immunoreactivities in pre-treatment primary tumors derived from NPC patients diagnosed with complete tumor remission, tumor relapse, or metastasis after treatment. Only those samples grouped in the “metastatic lesions” are tumor biopsies derived from metastatic sites (Fig. 1b-c). Results of statistical analysis indicate that higher LIF/LIFR expression in primary NPC tumor is a risk factor developing metastasis or recurrence (Table 1 and Table 2). Since it’s very difficult to obtain primary and metastatic biopsy pair from the same individual, we could not analyze the temporal change of LIF/LIFR in a limited sample size. The description of detailed immunohistochemical analysis has been added to the paragraph of “Immunohistochemistry” in the section of Methods (page 33).

2. The use of TALENs to mutate the signal peptide region of LIF is an interesting approach to study the cytoplasmic function of LIF. But I remain concerned that the cytoplasmic staining observed in patient samples could simply represent a surrogate marker of cells that produce and secrete abundant levels of LIF. This would seem to be consistent with the prognostic significance of LIFR that is reported here. To put it another way, are the engineered LIFc clones faithful representatives of the NPC carcinoma cells observed in Figure 1? Given that there is no evidence for this type of signal peptide mutation in LIF, the LIFc clones could be rather idiosyncratic.

Our response:

Similar question to Referee 1, question #2; also the very first question of reviewer #2)

The TALEN technology is a tool to study the potential mechanisms underlying our observations. Also, whether other SNVs or mutations are associated with LIF/LIFR

expression will need further experimental and clinical validations.

3. There are a number of problems with the conclusions drawn at the end the section titled “Characterization of established LIF mutant cancer clones.” For one, it states that the LIFc cells are not desensitized by exogenous LIF. It would be better, and more accurate, to simply state that they are not responsive to exogenous LIF. Second, it states that “these findings clearly indicate that cytoplasmic LIF and LIFR are co-expressed,” which doesn’t seem like the right conclusion here. Do the authors mean that cLIF can regulate the expression of LIFR? Or that the cLIF is functional in the engineered clones?

Our response:

- (1) We have corrected the sentence as suggested (page 12). Thank you.
- (2) More accurately, our data suggested that the LIF can regulate LIFR expression. We have revised our sentence as “These findings indicate that LIF can regulate LIFR expression " (page 12). Thank you.

4. In regard to the vascular invasion assays where LIFc cells are able to disrupt the HUVEC layer, are the authors sure that no LIF is secreted from the LIFc cells? In general this seems like an important control to include, but especially so here in order to ensure that secreted LIF is not acting directly on the endothelial cells. Also, do LIFc cells exhibit enhanced migration in a transendothelial migration assay?

Our response:

- (1) We thanks for referee’s suggestion. We have utilized a sensitive bead-based cytokine assay to detect the levels of supernatant LIF secreted by these three cell lines. As shown Fig. 2c, the WT cells secreted higher amount of LIF (mean \pm SEM (pg/ml) : 37.7 ± 1.3) into medium whereas the LIFc cells secreted little amount of LIF (6.9 ± 1.5) into culture supernatant. The LIF level secreted by the LIF^{+/-} cells is about half of the WT (17.6 ± 1.6). The experimental procedure has been added to the Supplementary information and the results were incorporated into the revised manuscript (page 11, Fig.2c).

Fig. 2c

Fig.2c Assessment of secreted LIF using a bead-based cytokine assay. Supernatants were harvested 2 days post culture. ****** $p < 0.01$, two-tailed, paired t test.

(2) Indeed, the LIFc cells exhibit an enhanced transendothelial migration ability. They may invade the endothelial layer via intercellular migration or paracellular migration. We frequently observed that LIFc cells could invade the HUVEC layer and colonize at the inner side of endothelial layer (see Ref Figure1 below).

Ref Figure 1

Ref Figure1. The endothelial invasion patterns of LIFc cells. **a.** LIFc cells directly caused HUVEC damage and generated a hole on the HUVEC layer. **b.** LIFc cell

protrusions invaded into the HUVEC layer. **c, d.** LIFc cells colonized at the inner side of HUVEC layer. Cancer cells were plated on top of the confluent HUVEC layer and co-cultivated for 24 h. Cells were fixed and labeled with antibodies against pan-cytokeratin (red) and VE-cadherin. Scale bar, 20 μ m.

5. The western blots are confusing in these co-culture assays. Do these lysates represent bulk lysates of HUVEC and BM1 cells?

Our response:

The results shown in Fig. 8e are lysates of HUVEC cells plus BM1 cancer cells. To make it clear, we have revised the description of this experiments (Fig. 8e and the Figure legend).

6. Using the zebrafish xenograft model the authors state the LIF is critical for “cancer vascular evasion.” To clarify, in this model does this represent intravasation or extravasation?

Our response:

We used the zebrafish model to evaluate the cancer vascular dissemination by counting the numbers of tumor-like structures along the fish vessels. We currently are unable to distinguish the events between intravasation and extravasation from this model due to the microscopic resolution.

7. The authors argue that LIFc switches the mode of invasion from collective to mesenchymal (single-cell). The evidence for this comes largely from a wound-healing assay. Can a similar behavior be observed using 3D assays that have been used to study this behavior (see Cheung, Cell, 2014; Ye, Nature, 2015)?

Our response:

Thanks for the constructive suggestion. We have conducted the 3D matrix invasion assays to demonstrate that the LIFc cells are more invasive on the 3D matrix model. Cells were plated on the surface of 35 mm dishes coated with stiff gelatin matrix (6% gelatin/PBS, the height of matrix is ~160 μ m, the pressure for 6% gelatin is about 30 kPa). The invasion depth and distance were continuously measured by using time-lapse phase contrast vertical scanning (Olympus IX83). Results demonstrated that LIFc cells could invade into deeper depth compared with WT or LIF^{+/-} cells (38 μ m vs. 10 μ m after 5 hours post plating (page 13, Fig. 3b). Further, the change of invasion mode from collective to mesenchymal was supported by (1) wound-healing assay (Fig. 3a, Supplementary video 1-3); (2) expression of EMT markers (Fig. 3e); (3)

morphological changes and cytoskeletal reorganization (Fig. 3f-h and Fig. 5b).

Fig.3b

Fig.3b 3D-matrix invasion assays. The invasion depth of cancer cells through a stiff matrix were measured by time-lapse phase contrast vertical scanning (Olympus IX83).

8. The weakest aspect of this study involves the connection between LIF, LIFR and the YAP pathway. There are a multitude of issues here that make interpretation of the results difficult. 1. LIFc/LIFR-high cells have low YAP1 levels but knockdown of LIFR leads to increased phosphorylation of YAP. These two scenarios (p-YAP vs. total YAP) are not equivalent as the authors seemingly argue. 2. There is also the suggestion that LIFc cells have lower p-YAP but relative to total YAP this doesn't seem to be true. 3. The changes in p-PXN and p-FAK after YAP knockdown aren't very convincing. Thus, with the data presented it is very difficult to support the mechanistic model they have put forth where LIFc inactivates YAP. What happens if a constitutively active YAP is expressed in LIFc cells? And does their model support the idea that YAP plays some sort of functional role in cells with low LIF/LIFR?

Our response:

We have conducted more experiments to answer referee's comments.

(1) In our cell model, LIF/LIFR-rich cancer cells (LIFc) expressed lower levels of phosphorylated YAP1 (S127) as well as total YAP1 protein (Fig. 6a). From the staining results, YAP1 was predominantly expressed in the nucleus (Fig. 6b). But even so, the nuclear YAP1 expressed in LIFc remains lower than that observed in either WT or LIF^{+/-} cells (Fig. 6b). Knocking down LIFR increased level of YAP1 and p-YAPs (Fig. 6c), which suggests more YAP1 protein was trapped in the cytoplasmic compartment and that might contribute to the increased level of total YAP1. The regulation of YAP1 localization is critical for maintaining cell-cell junctions and mechanotransduction³¹. We currently do not have answer to

how LIFR regulates YAP1 expression. One potential explanation is that the invasive behavior of LIFc cells could be the consequence resulting from imbalanced N/C YAP1 distribution.

(2) **Partly similar as our answer in question #8 (1).**

The LIFc cells expressed lower levels of phosphorylated YAP1 (S127) as well as total YAP1 protein (Fig. 6a-b). We loaded equal amount of total protein in the western blotting as reflected by the protein level of GAPDH internal control. It is reasonably to describe that the p-YAP1 is lower in LIFc cells compared with WT and LIF^{+/-} cells.

(3) We have conducted repeated experiments and obtained consistent results demonstrating that depletion of YAP1 led to increased expression of p-PXN (Y118) and p-FAK (Y397) (Ref Figure 2). We have replaced the western blot with a more representative blot in Fig. 6d.

In addition, we've tried to express constitutively active YAP1 (127A) in these three cell lines. Ectopic expression of constitutively active YAP1 (S127A) in WT and LIF^{+/-} cells caused morphological change one day after transfection. However, cells started to undergo apoptosis approximately 2 days after transfection (Ref Figure 3, right panels). Most cells died out on the 3rd day. The phenomenon remained similar even we reduced the amount of plasmid DNA to as low as 0.3 ug/well (for 6-well plate). The LIFc cells appeared more sensitive to the expression of YAP1(S127A). Cell death was observed at 20~24 hours post transfection. Since the S127A YAP1 were predominantly localized in the nucleus of cancer cells, we speculated that the imbalanced expression of nuclear YAP1 was toxic to NPC cells.

Ref Figure 2

Ref Figure 2. Western blot analysis of expression of focal adhesion molecules in WT and LIF^{+/-} cancer cells transfected with YAP1 siRNA. GAPDH was used as the

loading control.

Ref Figure 3

Ref Figure 3. Ectopic expression of constitutively active YAP1 (S127A) in WT and LIF^{+/-} cancer cells. **a** Western blotting of ectopic expression of YAP1 (S127A) expression. **b** Immunostaining for YAP1 expression in WT and LIF^{+/-} cancer cells transfected with pcDNA4_YAP1_S127A plasmid or pcDNA4 control vector. Alexa Fluor 488 phalloidin (green) was used to stain F-actin. Blue, nuclear staining. Scale bar, 10 μ m.

9. The authors describe a correlation between LIFR expression and cytoplasmic YAP in clinical NPC samples (Fig. 6c) but in the LIFc cells YAP seems to exhibit a nuclear localization (Fig. 7b). How do the authors reconcile this discrepancy?

Our response:

Partly similar as our answer in question #8.

In our observation, both p-YAP1 (S127) and total YAP1 are lower in LIFc cells compared with that in WT or LIF^{+/-} cells. Expression of YAP1 was found in compartments of cytosol and nucleus in WT and LIF^{+/-} cells but predominantly localized in the nucleus of LIFc cells (Fig. 6a-b, Fig. 7b). The nuclear YAP1 expression in LIFc cells appears to be lower than that observed in WT or LIF^{+/-} cells. Since the regulation of YAP1 localization is critical for maintaining density-dependent cell-cell junctions (cytoplasmic YAP1) and mechanotransduction (nuclear YAP1)³¹. We speculate that the imbalanced nuclear/cytosolic YAP1 expression attributes to the

invasive phenotype of LIFc cells.

10. The use of AZD0530, although it has some effects, seems somewhat arbitrary. Do LIFc cells have higher levels of active Src? Is Src downstream/upstream of LIFc/LIFR or does it function in a parallel pathway. Does AZD0530 have any effect on the viability/proliferation of the LIFc cells? How does this compound affect the LIFwt and LIF^{+/-} cells? Some basic controls are needed here.

Our response:

The LIFc cells express higher levels of activated SRC (Fig. 3e & Fig. 6c-d). Results from the knockdown experiments, both LIFR and YAP1 can regulate expression of p-SRC (Y416) (Fig. 6c-d). We have conducted the survival assays to evaluate the toxicity of each cell lines in response to AZD0530 treatment. Results showed that the LIFc cells were more resistant to AZD0530 treatment ($IC_{50} = 1.73 \times 10^{-5}$ M) whereas the LIF^{+/-} cells were more sensitive to AZD0530 treatment compared with LIF wild-type cells ($IC_{50} = 8.93 \times 10^{-7}$ M). We have revised our description about the use of AZD0530 in this study (page 21-22, Supplementary Fig. S5).

Supplementary Figure S5

Supplementary Fig. S5 Assessment of the IC₅₀ values of AZD0530 for WT and LIF mutant cancer cells. **a, c, e** Real-time measurement of cell survival in response to various doses of AZD0530 treatment. Values of cell survival were normalized with respect to the time of AZD0530 addition. Values are presented as means and SD of quadruplicate experiments. **b, d, f** The AZD0530-induced cell toxicity was determined by calculating the IC₅₀ after treatment. Normalized cell index was displayed against the logarithm of concentration. The values of IC₅₀ were calculated using the RTCA software (ACEA Biosciences).

11. The mechanism put forth by the authors is rather complicated and it seems a simpler interpretation could explain their findings. In Figure 2 and 3 there is pretty clear evidence that the LIFc cells have undergone an EMT. The EMT program has been shown to regulate tumor cell invasion (Nieto, Cell, 2016), invadopodia formation (Eckert, Cancer Cell, 2011), transendothelial cell migration (Drake, MBoC, 2009), and the expression of focal adhesion molecules (Shibue, Cancer Cell, 2013). Thus, it seems that many, if not all, of the observations reported here can be explained by the fact that LIFc cells are more mesenchymal than the LIFwt and LIF+/- cells.

Our response:

Thanks for the suggestion. In this study, we found that increased cellular LIF promotes invadopodia formation, EMT process, and cancer dissemination by suppressing phosphorylation of YAP1 (S127) via LIFR, and activating SRC signaling and focal adhesion assembly. Based on our findings, we propose a graphic abstract in this revised version. It is presented on page 24 and Figure 9.

Fig. 9 The schematic model for LIF-mediated cancer dissemination in NPC.

12. References 12 and 19 are the same.

Our response:

We have deleted reference 19, thank you.

References

1. Cerami, E., *et al.* The cBio cancer genomics portal: an open platform for exploring multidimensional cancer genomics data. *Cancer Discov* **2**, 401-404 (2012).
2. Gao, J., *et al.* Integrative analysis of complex cancer genomics and clinical profiles using the cBioPortal. *Sci Signal* **6**, p11 (2013).
3. Rhodes, D.R., *et al.* OncoPrint 3.0: genes, pathways, and networks in a collection of 18,000 cancer gene expression profiles. *Neoplasia* **9**, 166-180 (2007).
4. Cerami, E., *et al.* The cBio cancer genomics portal: an open platform for exploring multidimensional cancer genomics data. *Cancer discovery* **2**, 401-404 (2012).
5. Reinartz, S., *et al.* A transcriptome-based global map of signaling pathways in the ovarian cancer microenvironment associated with clinical outcome. *Genome Biol* **17**, 108 (2016).
6. Liu, S.C., *et al.* Leukemia inhibitory factor promotes nasopharyngeal carcinoma progression and radioresistance. *J Clin Invest* **123**, 5269-5283 (2013).
7. Chen, D., *et al.* LIFR is a breast cancer metastasis suppressor upstream of the Hippo-YAP pathway and a prognostic marker. *Nature medicine* **18**, 1511-1517 (2012).
8. Okamura, Y., *et al.* Leukemia inhibitory factor receptor (LIFR) is detected as a novel suppressor gene of hepatocellular carcinoma using double-combination array. *Cancer Lett* **289**, 170-177 (2010).
9. Zeng, H., *et al.* Feedback Activation of Leukemia Inhibitory Factor Receptor Limits Response to Histone Deacetylase Inhibitors in Breast Cancer. *Cancer Cell* **30**, 459-473 (2016).
10. Pan, D. The hippo signaling pathway in development and cancer. *Dev Cell* **19**, 491-505 (2010).
11. Zeng, Q. & Hong, W. The emerging role of the hippo pathway in cell contact inhibition, organ size control, and cancer development in mammals. *Cancer Cell* **13**, 188-192 (2008).
12. Yuan, M., *et al.* Yes-associated protein (YAP) functions as a tumor suppressor in breast. *Cell Death Differ* **15**, 1752-1759 (2008).

13. Hiemer, S.E., *et al.* A YAP/TAZ-Regulated Molecular Signature Is Associated with Oral Squamous Cell Carcinoma. *Mol Cancer Res* **13**, 957-968 (2015).
14. Zhang, J.G., *et al.* Identification and characterization of two distinct truncated forms of gp130 and a soluble form of leukemia inhibitory factor receptor alpha-chain in normal human urine and plasma. *J Biol Chem* **273**, 10798-10805 (1998).
15. Auernhammer, C.J. & Melmed, S. Leukemia-inhibitory factor-neuroimmune modulator of endocrine function. *Endocr Rev* **21**, 313-345 (2000).
16. Tomida, M. Structural and functional studies on the leukemia inhibitory factor receptor (LIF-R): gene and soluble form of LIF-R, and cytoplasmic domain of LIF-R required for differentiation and growth arrest of myeloid leukemic cells. *Leukemia & lymphoma* **37**, 517-525 (2000).
17. Pitard, V., *et al.* The presence in human serum of a circulating soluble leukemia inhibitory factor receptor (sgp190) and its evolution during pregnancy. *European cytokine network* **9**, 599-605 (1998).
18. Zhang, J.G., *et al.* Identification and characterization of two distinct truncated forms of gp130 and a soluble form of leukemia inhibitory factor receptor alpha-chain in normal human urine and plasma. *The Journal of biological chemistry* **273**, 10798-10805 (1998).
19. Owczarek, C.M., Layton, M.J., Robb, L.G., Nicola, N.A. & Begley, C.G. Molecular basis of the soluble and membrane-bound forms of the murine leukemia inhibitory factor receptor alpha-chain. Expression in normal, gestating, and leukemia inhibitory factor nullizygous mice. *The Journal of biological chemistry* **271**, 5495-5504 (1996).
20. Elsaid, M.F., *et al.* Non-truncating LIFR mutation: causal for prominent congenital pain insensitivity phenotype with progressive vertebral destruction? *Clin Genet* **89**, 210-216 (2016).
21. Guran, T., *et al.* Effects of leukemia inhibitory receptor gene mutations on human hypothalamo-pituitary-adrenal function. *Pituitary* **18**, 456-460 (2015).
22. Mikelonis, D., Jorcyk, C.L., Tawara, K. & Oxford, J.T. Stuve-Wiedemann syndrome: LIFR and associated cytokines in clinical course and etiology. *Orphanet J Rare Dis* **9**, 34 (2014).
23. Haines, B.P., Voyle, R.B., Pelton, T.A., Forrest, R. & Rathjen, P.D. Complex conserved organization of the mammalian leukemia inhibitory factor gene: regulated expression of intracellular and extracellular cytokines. *Journal of immunology* **162**, 4637-4646 (1999).
24. Haines, B.P., Voyle, R.B. & Rathjen, P.D. Intracellular and extracellular leukemia inhibitory factor proteins have different cellular activities that are

- mediated by distinct protein motifs. *Mol Biol Cell* **11**, 1369-1383 (2000).
25. Hisaka, T., *et al.* Expression of leukemia inhibitory factor (LIF) and its receptor gp190 in human liver and in cultured human liver myofibroblasts. Cloning of new isoforms of LIF mRNA. *Comp Hepatol* **3**, 10 (2004).
 26. Guo, H., Cheng, Y., Martinka, M. & McElwee, K. High LIFr expression stimulates melanoma cell migration and is associated with unfavorable prognosis in melanoma. *Oncotarget* **6**, 25484-25498 (2015).
 27. Kuphal, S., Wallner, S. & Bosserhoff, A.K. Impact of LIF (leukemia inhibitory factor) expression in malignant melanoma. *Exp Mol Pathol* **95**, 156-165 (2013).
 28. Maruta, S., *et al.* A role for leukemia inhibitory factor in melanoma-induced bone metastasis. *Clin Exp Metastasis* **26**, 133-141 (2009).
 29. Khosla, S. Minireview: the OPG/RANKL/RANK system. *Endocrinology* **142**, 5050-5055 (2001).
 30. Hughes, A.E., *et al.* Mutations in TNFRSF11A, affecting the signal peptide of RANK, cause familial expansile osteolysis. *Nature genetics* **24**, 45-48 (2000).
 31. Elosegui-Artola, A., *et al.* Mechanical regulation of a molecular clutch defines force transmission and transduction in response to matrix rigidity. *Nat Cell Biol* **18**, 540-548 (2016).

Point-by-point responses to reviewers' comments

Reviewer #1

The authors addressed very well all my queries in the revised manuscript.

Reviewer #3

The manuscript by Liu and colleagues examines the role for cytoplasmic LIF/LIFR signaling in NPC tumor progression. The manuscript was very well written, and the experiments were conducted with appropriate controls. The authors appropriately highlight the complexity and tumor-specific differences in the role for LIF/LIFR. One issue should be addressed:

The authors state in Figure 6 that there are changes in pYAP in LIFc and siLIFR cells, but there are also changes in total YAP as well. Please quantify the changes in pYAP and total YAP in Figure 6a and 6c. If pYAP is not in fact elevated when normalized to total YAP, this statement should be corrected.

Our Response:

Thanks for the suggestion, we have followed the suggestion to quantify the changes in p-YAP and total YAP presented in Figure 6a and 6c. The levels of p-YAP1 and total YAP1 were first normalized to the levels of loading control, and the changes in p-YAP1 were then further normalized to the levels of total YAP1. We have revised our description about the increased phosphorylation of YAP1 (S127) by LIFR siRNA in cLIF cells (Figure 6c) (page 19, line 7-13). The blots have been revised as below:

Figure 6 LIFR-YAP1 signaling is critical for LIF-mediated invasion of NPC cells. **a** Endogenous protein expression of LIFR, p-YAP1(S127), and YAP1 in three cancer cell lines. **c** Western blot analysis of LIFR and p-YAP1 (S127) protein levels in WT and cLIF cancer cells transfected with SMARTpool LIFR siRNA or control siRNA.